# Participatory Urban Design for Touristic Presentation of Cultural Heritage Sites: The Case of Negotinske Pivnice (Wine Cellars) in Serbia

Zoran Đukanović , Jelena Živković , Uroš Radosavljević , Ksenija Lalović and Predrag Jovanović *

Department of Urbanism, Faculty of Architecture, University of Belgrade, Bulevar kralja Aleksandra 73/II, 11000 Belgrade, Serbia; duke@arh.bg.ac.rs (Z.Đ.); jelena.zivkovic@arh.bg.ac.rs (J.Ž.); yros@arh.bg.ac.rs (U.R.); ksenija.lalovic@arh.bg.ac.rs (K.L.)
* Correspondence: predrag.jovanovic@arh.bg.ac.rs

**Abstract:** The growing perception of heritage as a public commodity encourages rural communities to recognize their natural and cultural heritage as a potential for tourism development. This creates the need for an appropriate presentation of heritage sites that ensures that their cultural and natural assets are appreciated and protected. "Negotinske pivnice" are cultural heritage sites in Serbia, nominated for the UNESCO World Heritage List. They are architectural complexes of wine cellars in rural region with a long tradition in wine production and industry and are unique in terms of their settlement structure. This emphasizes the spatial dimension of their interpretation and presentation, and highlights the importance of urban design for their sustainable use for tourism. Based on understanding urban design both as a process and a product, and cultural heritage site as a place, we argue that participatory urban design contributes to appropriate heritage presentation by widening design knowledge base to include local communities' lay knowledge. Following the case study methodology, we explored the relationships between participatory process, the knowledge gained, and urban design solutions for presentation of cultural heritage sites as living places in "Wine Cellars of Negotin Participatory Urban Design" project. The research reveals that the wider knowledge base affects urban design at both strategic and project levels and sets the grounds for diverse presentation forms through which harmonization of heritage protection and touristic presentation is possible.

**Keywords:** presentation of cultural heritage; tourism; urban design; participation; wine cellars





## 1. Introduction

In the context of growing use of cultural heritage for tourism, there is a need for the appropriate use, interpretation and presentation of heritage sites, which ensures that local people are involved and benefit from their development, and that local cultural and natural assets are respected and protected.

The purpose of this paper is to discuss community participation in urban design for presentation of cultural heritage sites (CHS), as a possible way to harmonize heritage conservation and tourism development in rural setting. This research study is situated within the rural context of the Eastern Serbia and focuses on the investigation of the participatory urban design process and the results of that process, in order to reveal how can local communities' knowledge guide urban design for an appropriate presentation of CHS. A representative case study of "Wine cellars of Negotin—participatory urban design" (WCN PUD) project is presented and analyzed as an example of a successful practice.

### 1.1. Inclusive Approach to Heritage Management for Sustainable Tourism

Under the new paradigm [1–3], understanding and management of heritage change. Traditionally, heritage properties included individual monuments, places of worship, fortifications or buildings, without taking into account their relationship with the surrounding

landscape. Today, it has been recognized that the "whole environment is shaped and affected by its interaction with humanity and can be recognized as a heritage" [4]. Consequently, it becomes necessary to judge about what is considered as heritage. The concept of heritage is assumed to be culturally constructed, and that there is an almost infinite variety of possible "heritages" [5]. In that sense, history and heritage are no more only the domain of academics and specialists, but are recognized as the resources for community and economic development, medium for cultural identity, a destination for cultural tourists, and a basis of educational enrichment [6]. As such, heritage is no longer passively protected, but is now perceived as a part of development [2,3], and linked to social, economic and environmental sustainability [4].

In that context, cultural heritage sites (CHS) are understood as *places* [7] and not mere physical structures. The concept of *heritage places* recognizes heritage as a key anchor for cultural identity and therefore an important element of community development [1]. Araoz [1] suggests that "the range of values attributed to heritage places has expanded to reflect its new social role as well as the many ways in which it is appreciated by stakeholding communities whose voices had not been given major consideration in the past". As such, CHS are conceived as both powerful educational instruments and means of sustainable community development that cannot be protected in isolation from natural and cultural impacts, social changes, planning and development considerations, or separated from the concerns of the communities [4].

All of this is especially important for the development of rural communities that recognize their natural and cultural heritage as a potential for tourism development. In parallel, new trends in demand and the rise of cultural tourism, affect tourist geographies and widen the spectrum of places and activities that are of interest to contemporary tourists. In this context, even remote rural regions that are rich in natural and cultural heritage have an opportunity to prosper from tourism.

Rural heritage used to be narrowly defined and considered to consist of buildings associated with agricultural activity, wash-houses, mills or chapels. Today, planners are assigning a broader definition of rural heritage, considering that it includes "all the tangible or intangible elements that demonstrate the particular relationship that a human community has established with a territory over time" [8]. It consists of the landscapes shaped through time by people who lived off the land, the buildings that make up what is referred to as rural architecture, but also of the local products that reflect specific local conditions and needs, as well as the techniques, tools and know-how [8].

However, the use of cultural heritage for tourism development is also criticized in literature for possible negative effects that it brings to heritage sites and to local communities. It is suggested that the transformation of cultural heritage sites into venues for tourism poses great threats to their conservation, and that if the right balance between different stakeholders is not achieved, "the site development project, even if financially successful, can appear to local residents as an outside imposition—like a shopping mall or private theme park—with solely or mainly economic significance for the community" [9].

There is therefore a growing need for the appropriate use of heritage sites for tourism, which ensures that local people are involved and benefit from their development, and that local cultural and natural assets are appreciated and protected. It is the community of local people, who perform their daily activities, and produce and reproduce their social relations and culture, that give value and meaning to a heritage site [10], and should therefore be included in its development and management.

In that context, the concept of "sustainable tourism" is developed, referring to "tourism that takes full account of its current and future economic, social and environmental impacts, addressing the needs of visitors, the industry, the environment and host communities" [11]. In this view, it is assumed that the involvement and co-operation of local and/or indigenous communities, conservationists, tourism operators, policy makers, planners and site managers are necessary for both sustainable tourism development and for the protection and enhancement of heritage resources [12].

In this more complex development situation, heritage management is no longer seen as only the experts domain but involves different stakeholders. It also assumes different ways of knowing heritage places and appreciation of lay knowledge and subjective perception of heritage sites. In that context, special attention is now paid to the inclusion of local communities in all stages and aspects of heritage management. Therefore, a more inclusive approach is promoted in order to better balance and harmonize heritage protection, tourism development and local quality of life [13–16]. Involving citizens and stakeholders in planning and development process is based on the assumption that "participation is right to take part in the local governance, a process that allows influence and control over decision making, but also a process of learning how to listen, recognize and accept different opinions, feelings, values and knowledge." [17].

In relation to this, contemporary heritage management models evolve. Today, three dominant approaches exist [18]. "The Authorized Heritage Discourse" as the model of heritage preservation "focuses attention on aesthetically pleasing material objects, sites, and places and/or landscapes that current generations 'must' care for, protect and revere' on behalf of the public" [19]. "Values based" approach promotes the idea that stakeholders ascribe value to sites, and that the main reason why heritage is conserved is not for the sake of the material but because of the values ascribed to it [7]. This approach does not diminish the value of the physical materials but highlights the importance of the conservation of other tangible and intangible elements, such as the conservation of landscape views and traditional uses [20]. More recently a new concept of "living heritage" is developed, that directly builds upon the active involvement of local communities in heritage management. It refers to sites, traditions, and practices developed over time by many people, and which are still in use. The key concept in defining living heritage site is that of *continuity*: of the function of a site, of the process of maintenance and of the physical presence of a site's community in a site [21]. It is assumed that the people who live, work or visit the site daily are the most responsible for the heritage maintenance, and are key-actors that enable its sustainable grassroots conservation [22].

There is growing research on community involvement and participation in urban and heritage planning and management, as well as in planning for sustainable tourism. In urban and spatial planning literature, debates on inclusiveness and participation have focused on three main aspects: the diversity of representation of local communities, the level of participation and the timing of public engagement [23]. Besides that, it has been recognized that different forms of community involvement in planning and development may exist, such as: informing, consulting and active participation [17]. In the heritage management field, Luna Khirfan [23] provides detailed analysis of the evolving use of public participation in heritage planning process, based on analysis of content of relevant UNESCO and ICOMOS heritage documents. Apart from that, other authors provided valuable research on individual cases in relation to this topic [10,20,24,25]. In tourism literature, the research has mainly been focused on examples of good practice [26–28] resulting in identification of key factors contributing to successful practice. Although stemming from different lines of research, the literature shows that the involvement of local communities in the development process can be best understood when related to *stakeholders/actors, phases and forms* of participation, and the research of successful community participation highlight the importance of *diversity and continuity* in participation process and heritage planning and management.

But, although community involvement and participation have been researched in relation to urban, heritage and tourism planning and management in general, little is known about how it can contribute to the development and presentation of rural heritage sites [25]. On the other hand, inclusiveness in interpretation and presentation are recognized as important for CHS conservation and management, and are of special importance in its growing use for tourism. Silberman [29] suggests that "For even if cultural heritage sites are interpreted in the widest possible environmental and historical contexts, they can still be regarded by modern visitors and residents as isolated enclaves, intentionally

taken out of place and time. This is true not only for discrete monuments with perimeter fences and ticket booths, but also for historic town centers and cultural landscapes if they are too consciously set apart from daily life." In that sense, the inclusive approach to heritage management, interpretation and presentation, and the input and involvement of local and associated community groups, visitors and other stakeholders, are essential for transforming cultural heritage sites from static monuments into places of learning on past, intercultural dialogue and valuable resources for sustainable community development [9].

### 1.2. Presentation of Cultural Heritage Sites

Heritage interpretation and presentation are part of the overall heritage management system, and according to benefits they can provide, they may be as important as the physical preservation of a site. Their purpose is to communicate cultural values and identities, to enhance visitors' understanding and enjoyment of, and to help develop a positive attitude toward the heritage site [30], to increase public awareness and acquire support in the activities directed at its management and preservation. It has been assumed that quality heritage interpretation and presentation can bring a wide spectrum of benefits, such as promotional, educational, conservation /management and economic benefits [31–35].

The ICOMOS Charter for the Interpretation and Presentation of Cultural Heritage Sites (2008)—Ename Charter [35] is an important document that aims to guide quality interpretation and presentation of cultural heritage sites. It recognizes the difference between interpretation and presentation and stresses that interpretation refers to the full range of potential activities, while presentation "more specifically denotes the carefully planned communication of interpretive content through the arrangement of interpretive information, physical access, and interpretive infrastructure at a cultural heritage site", and can be conveyed through a variety of technical means, such as: informational panels, museum-type displays, formalized walking and guided tours, lectures and multimedia applications and websites [35].

The main aim of the ICOMOS—Ename Charter is to "ensure that every community's interpretation of its cultural heritage sites is inclusive, authentic, sustainable, and—yes—an endless source of knowledge, inspiration, and reflection about the past's evocative, enigmatic, and always enlightening material remains." [9]. The Charter [35] defines the basic objectives and principles of site interpretation and presentation in relation to authenticity, intellectual integrity, social responsibility, and respect for cultural significance and context. These principles emphasize the essential roles of public communication and education in heritage preservation. They are presented under the headings: Access and Understanding; Information Sources; Context and Setting; Authenticity; Sustainability; Inclusiveness; and Research, Education, and Training.

Besides the planned CHS presentation (*formal*), the everyday use of CHS by local people is also a form of their (*informal*) presentation as being part of tourist interest and curiosity. The authors who investigated the relationship between cultural heritage and tourism suggest that in using cultural heritage for tourism, it is important to recognize that it is not only designated cultural heritage that attracts and motivates tourists to visit heritage sites. The key to effective heritage presentation and interpretation lies in understanding the motivation for the visit and the type of experience sought by visitors [26]. Motivations for visiting CHS vary among tourists [27,36]. Kerstetter et al. [37] recognize the continuum: from general leisure travellers seeking a relaxing leisure experience; via tourists interested in gaining knowledge of the past but for pleasure seeking motives; to heritage tourism specialists who are seeking knowledge from the past. Herbert's [28] conceptual model points to a circular process relating to the production and presentation of heritage sites to visitors by management and the *diverse* ways in which visitors interpret messages. In that context, the variety of presentation formats and opportunities for both formal and informal presentations of CHS are of crucial importance.

In addition, the approach to interpretation and presentation of CHS depends on the type of heritage and character of heritage site that is to be presented to the public. When

buildings and sites are heritage, scattered in urban or rural landscape, the design of public spaces aims to support their connection and to contain presentation infrastructure. But when the settlement itself is heritage, a more complex approach to presentation is needed, and all physical elements, that constitute the environment, enable activities and convey meanings, may become constitutive parts of heritage presentation [38]. Apart from that, rural vernacular heritage usually requires that besides presenting tangible elements of heritage, local residents' life-world is revealed in relation to the territory where it takes place [39]. In that context, settlement spaces should be understood and planned not only as containers of heritage presentation infrastructure, but as heritage themselves, and "fully integrated into the modern life of the community in such a way as to retain local practices and ways of life" [40]. It is specifically in these situations, that urban design becomes an important part of heritage presentation.

### *1.3. Participatory Urban Design*
#### 1.3.1. Urban Design as Process and Product

Urban design (UD) is a process and a product of designing and shaping the physical features of man-made environment by creating connections between people and places, nature and built fabric, movement and form [41]. It is concerned with settlements of all sizes, including villages and rural settings. Urban design deals with different elements of urban form and operates at different scales. Elements of urban form include: urban structure, urban grain, density + mix, height + massing, streetscape + landscape, façade + interface, details and materials [42]. Scales of urban design vary from the macro scale of the urban structure to the micro scale of street furniture and lighting [42].

Although, historically, there are two broad traditions of urban design, namely "visual-artistic" and "social usage" tradition, in recent years they became blended into a third—"making places" tradition. This urban design approach simultaneously refers to urban space as an aesthetic entity and as a behavioral setting [43]. The unique identity of place is understood as the result of layering of buildings and infrastructure, natural ecosystems, communities and cultures.

#### 1.3.2. Knowledge for Urban Design

Urban design builds upon prerequisites and knowledge base that help define the purpose and guide decisions in different phases of the design process. Prerequisites may be initiated by the client and defined by the natural and cultural conditions of the site [44]. Specific social, cultural, economic and environmental characteristics and relations of a site need to be recognized, so that urban designer can establish a knowledge base and a framework for the design, and incorporate elements into a project. Or, as Kevin Lynch suggests: "Every site, natural or man-made, is to some degree unique, a connected web of things and activities. That web imposes limitations and offers possibilities. Any plan, however radical, maintains some continuity with the preexisting locale. Understanding a locality demands time and effort." [45].

Urban design connects knowledge to action through a systematic process that is related to specific context [44]. According to Palazzo and Stainer [44], the design process has four key steps: (1) Defining the problem; (2) Developing a rationale that takes into account summary analysis on planning, economic and social context, built form, land use, movement, environment, public realm, and perceptual and cultural factors; (3) Summarizing development opportunities and constraints; (4) Conceptualizing and evaluating design options.

All these steps can be also viewed as opportunities to involve local communities in the urban design process and contribute to the development of urban design solutions.

#### 1.3.3. Participation in Urban Design

Participatory design is an approach to design that actively involves different stakeholders in the design process to create environments that are responsive to their diverse

needs. In the context of urban design, participation is usually related to the inhabitants and future users of space and is considered as one approach to place-making. It encompasses various forms of pubic involvement in project design processes (public surveys, presentations, workshops, design charrettes, etc.) in order to develop or test design options with users. The appropriate type of community participation depends on the scale and nature of the project, as well as on the social and political context of development [46].

In urban design theory, Kevin Lynch [47] identifies congruence, user responsibility and certainty as three dimensions of 'control' that contribute to understanding of how variations in user participation and control affect the liveability of the city. Congruence refers to the degree to which users control the place, and is therefore balanced by the idea of responsibility, while certainty relates to the extent to which people understand and feel secure with the system of environmental control. Christopher Alexander [48] also advocate for user participation in design based on the assumption that people tend to take responsibility for their place if they are personally interested or feel they own it.

Theoretical propositions of Lynch and Alexander on benefits from user participation, are supported by research systematized in the publication "The Value of Urban Design" [46]. This publication provides clear and strongly supported findings that: (1) users provide information essential to the design process; (2) user participation leads to improved 'fit' between the environment and user needs; (3) responsiveness to the public and user concerns assist project approval processes; (4) user participation builds stronger communities; (5) user participation enhances democracy. The important thing recognized in this book is that "successful user participation processes in urban design rely on and do not substitute for professional design and technical expertise" as well as that there are inherent risks in a participatory process [46]. It is therefore suggested that the public should be involved at their level of competence, participating according to their interests and what they know [49].

Participatory urban design should help gathering appropriate information about the factors most affecting user satisfaction, contribute to proper respect for these factors, and strive for their reflection in the design [50]. The result of a well-conceived and managed participatory urban design process is a better and more responsive design of the environment. The dialogue developed through participation contributes to better understanding between different stakeholders and designers, and increases potential for their commitment to spatial change or development [46].

Therefore, in the context of applying this approach to rural CHS and based on understanding urban design both as a process and a product, and cultural heritage site as place, we argue that participatory urban design may contribute to appropriate heritage presentation to tourists and visitors by widening knowledge base to include local communities' lay knowledge to guide design solutions.

*1.4. The Case of "Negotinske Pivnice"(Wine Cellars of Negotin), Serbia*

In order to explore how can community involvement in urban design process contribute to appropriate presentation of rural, settlement-like heritage sites to tourists and visitors, a representative case of "Wine cellars of Negotin—participatory urban design" (WCN PUD) project was studied in this paper as an example of a successful practice of use of community participation for CHS presentation through urban design, in the context of Serbia.

A unique example of in situ protection of rural vernacular heritage in Serbia are "Negotinske pivnice" (Wine cellars of Negotin—WCN), nominated in 2010 for its outstanding value for the inscription on the UNESCO World Heritage List [51]. They are architectural complexes of wine cellars typical for Negotinska Krajina, a remote rural region of Serbia, with a long tradition in wine production and industry. Unlike other traditional wine cellars in Serbia and worldwide, "Negotinske pivnice" are unique because of their settlement structure. This gives the special importance to the spatial dimension of their interpretation and presentation as a cultural heritage and highlights the importance of quality urban

design through which their sustainable use for tourism may be enabled and meaning and value as a heritage conveyed.

Municipality of Negotin initiated in May 2014 a project for the improvement of public spaces of WCN and invited the University of Belgrade—Faculty of Architecture (UBFA) to conduct this project. This was a great opportunity to conduct in Serbian context a research and action project for a CHS presentation in which participation of local communities will lead urban design solutions. The project was conceptualized and tailored in accordance with specific social and spatial context and limits and aimed not only to provide an appropriate WCN presentation but also to educate future professionals–students as junior researchers in this field. The results were highly valued by the local community and awarded by experts.

In that sense, the WCN PUD project can be considered as an example of good practice and analyzed in order to improve general understanding on how participation of local communities can help appropriate CHS presentation through urban design in rural setting. The context, process and the results of the WCN PUD project will be presented in detail, and further analyzed in order to reveal the relationships between participatory process, the knowledge gained, and urban design solutions for presentation of cultural heritage sites as living places.

## 2. Materials and Methods

### 2.1. Methodological Approach

In this study a qualitative case study research strategy has been applied in order to answer the main research question and help reveal factors that support effective use of community participation for an appropriate presentation of CHS. The case study is a research approach (preferred in humanistic and social sciences, and frequently used for participation studies) in which the subject of the research is studied in depth and within its real-life context as an example of a real live phenomenon [52]. According to Yin [53] a case study research is best applied when the research addresses descriptive or explanatory questions: i.e., what happened, how, and why? In our case, this research strategy was chosen, since the main goal of this study was to describe the concept and the process of participation of local communities in developing urban design spatial strategies and solutions for CHS presentation in a local context, and to analyze the results of this participatory process in relation to design solutions in order to reveal how can local knowledge and values guide urban design for the presentation of WCN as a living place.

The research aims to reveal the relationships between participation process, community knowledge, character urban design projects and possibilities for CHS presentation, and is framed by three main questions:

- How can local community participation contribute to the formation of the urban design knowledge base?
- How can local community knowledge and perception of WCN as a living place guide urban design and what kind of CHS presentation WCN PUD project enables?
- Are there differences between WCN locations in relation to their potential for CHS management and presentation?

### 2.2. Choice of the Case

The value of the case study and the generalizability of findings depend on the strategic selection of cases. In our study, Wine cellars of Negotin participatory urban design project (WCN PUD) was analyzed as a rare and unique example of involving local communities into urban design for presentation in Serbia.

This case was chosen for three reasons. Firstly, the WCN PUD project deals with the important, settlement like wine cellars of Negotin region, that is formally protected national heritage site in a remote rural region of Serbia, also included on the WHS Tentative List. Besides that, these wine cellars complexes exist in several villages, which provided the opportunity to compare the results of the investigation.

Secondly, the WCN PUD project was initiated by the local community. The Municipality of Negotin has recognized WCN as important for sustainable tourism and socio-economic development of the Municipality and based on previous good experiences in working with UBFA initiated this research and action project.

Thirdly, the results of the WCN PUD project are positively verified by local people (municipality and village inhabitants) as well as by the professional community (1st award on 28th "International Urban Planners Exhibition" in Niš, Serbia). In that sense, the WCN PUD project can be understood and analyzed as good practice, and lessons learned from this project can be used to advance inclusive CHS planning and sustainable tourism theory and inform related practice.

Besides all of that, WCN PUD project is an educational project that can provide valuable insights for professional practice, while at the same time educating future professionals for inclusive approach to CHS planning for sustainable tourism development.

### 2.3. Research Design

Based on Yin's typology [53], our research design is conceptualized as a revelatory single case study with embedded units of analysis. Yin suggests that every type of case study design will analyze contextual conditions in relation to the "case," but recognizes four types of case study research designs that reflect different design situations [53]. These are single- and multiple-case studies, and these two variants can have unitary or multiple units of analysis. The resulting research design types are: (a) single-case (holistic) designs, (b) single-case (embedded) designs, (c) multiple-case (holistic) designs, and (d) multiple-case (embedded) designs. He suggests the rationale for single-case designs when it represents: (1) the critical case, (2) an extreme or a unique case; (3) representative or typical case, (4) revelatory case or (5) the longitudinal case [53].

In the context of Serbia, where inclusive CHS planning is in its infancy, application of participatory approach to development of UD solutions for touristic presentation through WCN PUD project provides a rare and unique opportunity "to observe and analyse a phenomenon previously inaccessible to social science inquiry" [53] and as such can be categorized as a revelatory case. In addition, to produce value, case study method needs a revelatory case to be linked to theory building or hypothesis testing [54].

Besides that, since the WCN PUD project applied one general participatory process at different locations in Negotin municipality—this study was conceptualized as a single case-study (WCN PUD project) with embedded units of analysis. Four different Wine Cellars (WC) complexes with their related villages in Negotin region (Rajac, Rogljevo, Štubik and Smedovac) were chosen as units for analysis of the potentials and limits of the PUD for presentation of CHS as living heritage. This is in line with Yin's suggestion that this type of research design may occur when, within a single case, attention is also given to a subunit or subunits. In addition, multiple units of analysis add opportunities for extensive analysis and enrich insights into the single case.

This approach enables that, besides the general analysis of how the participatory process may affect urban design solutions, the comparison can be made of the results obtained at different locations within the same conceptual approach to participation, and the same (municipality) level of development. The results of the comparisons further help analytical generalizations and identification of specific micro-contextual factors that shape UD solutions at both strategic and project levels. For that reason, comparative study contains a detailed and coordinated description of each case in order to establish the basis for cross-case comparison. It involves the analysis and synthesis of the similarities, differences and patterns across cases that have a common focus [55].

### 2.4. Data Collection and Analysis

Besides offering depth and specificity, case study research supports the diversity in terms of methods of data collection and analytical techniques [56]. This qualitative research on forms and effects of local communities' participation in the urban design

process for heritage site presentation is based on the conceptualization and realization of the WCN PUD project and uses its database for analysis. It builds upon WCN PUD project documentation that consists of diverse data related both to process and design phase of the study, and includes: field-notes, photographs and video materials, surveys, questionnaires, written statements, project design and visualizations. Part of the documentation material is published in a book "Wine cellars of Negotin: participatory urban design" [57].

Detailed description of data collection protocols and content, research methods and results, related to the WCN PUD project are presented in the next Section 3.3. WCN PUD project—phases. Besides that, heritage and tourism planning literature review has been conducted in order to identify the research gap in present knowledge on the topic and set the ground for this research. Content analysis of urban design literature has been performed to define key concepts and relations between urban design and community participation and help create analytical framework. In addition, critical review of Serbian heritage and tourism planning literature has been performed in order to identify the presence of an inclusive approach to CHS planning, and how Wine cellars of Negotin have been conceptualized, planned and researched.

Content analysis of the WCN PUD process and project documentation has been conducted in order to define specific development context, and to identify and systematize character and content of local inputs to the urban design knowledge base in relation to urban design strategies and projects, as well as in relation to presentation formats and character.

Finally, in order to draw conclusions on similarities and differences of WCN PUD outcomes, comparison through "word table" [53] was used for analyzing the patterns of integration of local knowledge and values into urban design strategy and projects. It displays the data from the individual units of analysis according to uniform framework, capturing the findings of four cases in relation to set of different UD and presentation characteristic.

## 3. Results

### 3.1. WCN PUD Project Location and Background

3.1.1. The Municipality of Negotin Location and Resources for Tourism Development

The Municipality of Negotin is located on the Danube River in eastern Serbia, near the borders between Serbia, Romania and Bulgaria. The Municipality has 37,056 inhabitants, of which 16,882 inhabitants live in the town of Negotin, and consists of 39 villages with a total of 16,597 households (census 2021). High depopulation rate and aging population are the main demographic characteristics of Negotin that are related to low nativity and migrations (Figure 1a).

As a remote rural municipality (250 km from Belgrade, the capital of Serbia), it is rich in natural resources and has a long tradition of wine production. Besides that, other natural resources offer possibilities for development of rural, vine, ecological, and hunting tourism. For all these reasons, strategic and planning documents highlight the possibilities for tourism development in relation to wine and heritage tourism, as well as rural, excursion, event and hunting tourism. "Strategy of Sustainable Development" of the Municipality of Negotin [58] recognizes the improvement of the conditions for tourism development as one of strategic goals, which beside other activities, includes the enhancement of public spaces of wine cellars of Negotin, and the support to traditional wine production.

It is on that basis that, building on positive experiences in previous collaboration with UBFA, the Municipality of Negotin initiated the WCN PUD project.

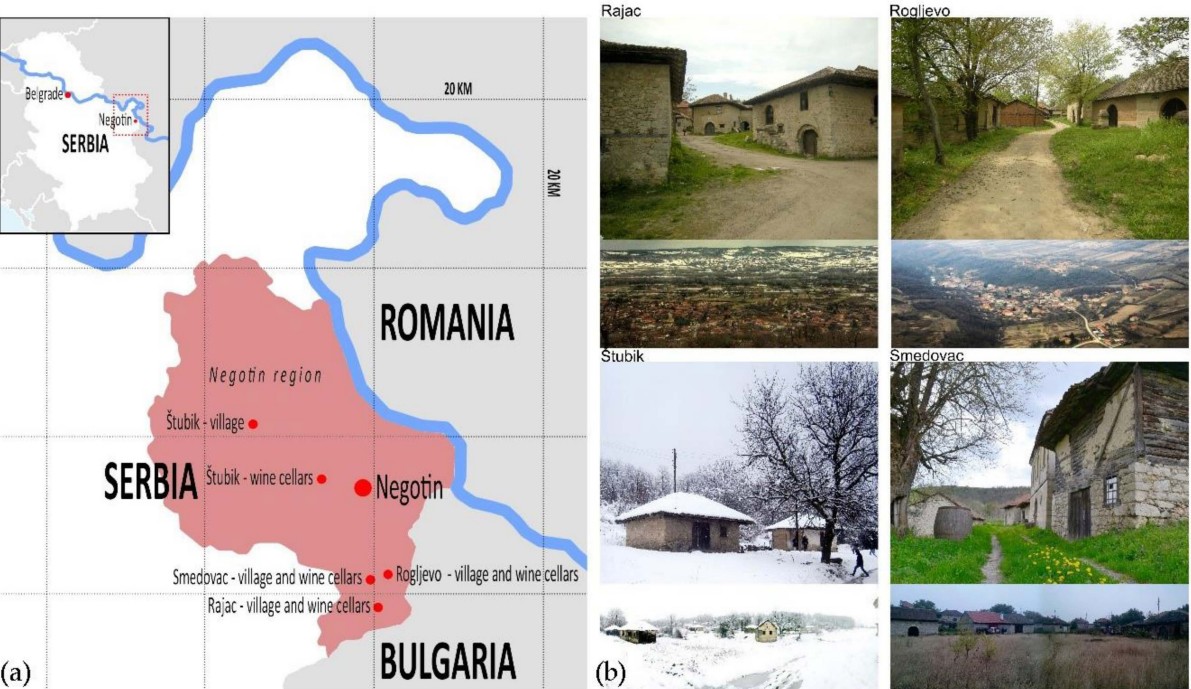

**Figure 1.** (**a**) The research area: Location of the Negotin region and wine-cellars; (**b**) Wine-cellars of Negotin: Rajac, Rogljevo, Stubik, Smedovac.

### 3.1.2. Wine Cellars of Negotin

"Negotinske pivnice" (wine cellars) are national cultural heritage site in Serbia, included in 2010 in the UNESCO World Heritage Sites tentative list due to its uniqueness, as well as authentic spatial and architectural values [59] (Figure 1b). They are architectural complexes of wine cellars typical for the area of the Negotinska Krajina (Negotin Frontier), famous for its vineyards dating from the ancient times. Traditionally, these rural compounds (settlements consisting of wine cellars) were built in the vicinity of vineyards as secondary settlements of rural communities and were used for making and storing wine and brandy. These settlements were named after the wine cellars and called "pivnice" [51]. They were mostly built of stone or woods and exemplify unique example of in-situ vernacular heritage in Serbia [59].

Although there were many in the past, today only wine cellar complexes in villages of Rajac, Rogljevo, Smedovac, Štubik and Bratujevo have been preserved, in different condition, use and level of protection. The Štubičke Pivnice and the Rajac cemetery were declared cultural heritage in 1980 and the Rogljevske Pivnice in 1983. In the same year, they were all classified as an area cultural-historic ensemble of outstanding value in the Republic of Serbia [51].

Nowadays, wine cellars are no longer being built, and in many villages these complexes collapse and decay. Those that remain active have been recognized by local communities as resources for tourism development. The hosts rearranged some of them into modern, functional spaces, retaining a specific ambience. This space is offered to tourists as small restaurants, shops or "private museums". Local people still gather in public spaces at special occasions, thus keeping and presenting local traditions.

The research on Negotinske pivnice is relatively scarce and three main lines can be distinguished. First one focuses on WCN as architectural heritage [60,61] and examines the application of specific design methods and customary rules in their construction and organization. Besides that, WCN have been partially explored in the field of tourism studies, with specific focus on wine tourism as a factor in the revitalization of rural settlements Rajac and Rogljevo [62] as well as how tourism development in Negotin region affects revitalization of traditional winemaking [63]. Importance of WCN for sustainable tourism

development has also been recognized in relation to rural tourism studies. The study "Integrated approach to sustainable development of rural tourism in Eastern Serbia" [64] specifically recognizes the need for the more inclusive approach to CHS management and tourism development.

Finally, an important studies of wine cellars and related villages have been conducted by the Institute for the Protection of Cultural Monuments of Serbia [65,66] and formed the base for development of Detailed regulation plans for Rogljevacke and Rajacke pivnice. In these studies, the tourist potential and management challenges of WCN have been recognized [59]. But, in these studies, traditional approach to cultural heritage dominates, although they recognize the importance of an integrated approach to tangible and intangible values and understanding of WCN in relation to village and landscape. What is missing is the recognition of local people's needs and perceptions of WCN as part of their everyday life. While the local people's initiatives have only been criticized for their inappropriateness, the solutions to problems of WCN management have been related to national authorities.

It is in this context that the WCN PUD research project was conceptualized and delivered, in order to fill this knowledge gap and contribute to the development of more inclusive CHS planning and management for sustainable tourism development in Serbia.

### 3.1.3. History of WCN PUD Project

WCN PUD project was initiated in May 2014 by the Municipality of Negotin and GIZ (Die Deutsche Gesellschaft fur Internationale Zusammenarbeit) and was realized as a part of an official collaboration between the Municipality of Negotin and the University of Belgrade—Faculty of Architecture.

This project builds upon their previous collaboration on several projects with strong participative dimension. The first one is an educational project (2011–2012) when in the context of Negotin region, two UBFA elective courses of "Public Art & Public Space—PaPs" program, engaged more than 100 students to research the relationship between wine, natural and cultural heritage development in order to propose public art interventions as support for development of rural economy in the area of Negotin municipality. The results were afterwards exhibited, published in the book "VinoGrad—The Art of Wine" [67], and some projects were implemented. The second one is a research project (2010–2012) conducted as a part of GIZ support to municipalities in Lower Danube region to jointly develop socio-economic projects for sustainable development of rural tourism. The Municipality of Negotin expressed the need to develop guidelines for sustainable design in authentic rural environments and for standardization of rural tourist accommodation. UBFA was chosen as a research institution, and the results were published in the study "Integrated approach to sustainable development of rural tourism in Eastern Serbia" [64]. Besides that, experiences of collaborative work with stakeholders in research and development of CHS in Negotin were presented in several publications [68–70].

The effective results and the success of the aforementioned projects, strengthened the relations of trust between partners, enabled different forms of learning, and formed the solid ground for future collaboration between the Municipality of Negotin and UBFA. On this basis, Municipality of Negotin, partnered with GIZ, initiated a new cycle of collaboration with UBFA in May 2014. In this new cycle UBFA had two important roles. The first one related to enabling the collaboration with Italian academics and municipalities in providing know-how and sharing experiences in development of similar rural regions, resulting in steps towards twinning with Italian municipality of Alghero, Sardinia. The other role of UBFA was research oriented and focused on exploring possibilities for enhancing public spaces of WCN in order to contribute to their sustainable revitalization, reconstruction, development and promotion as both CHS and tourist places. The research was partly conducted as an educational project of UBFA Master of architecture "Participatory urban design Studio" course in 2015–2016 (Figure 2).

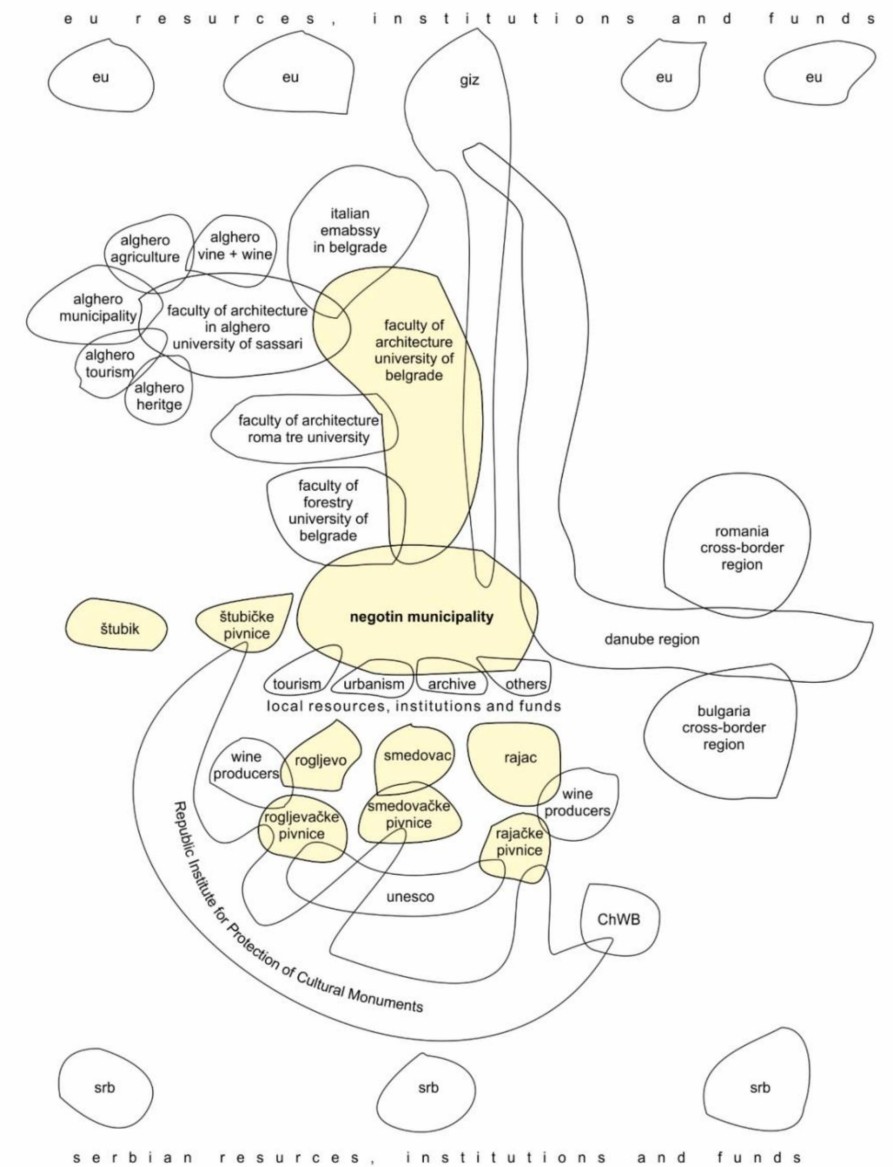

**Figure 2.** Context for Wine cellars of Negotin Participatory Urban Design (WCN PUD) project.

### *3.2. WCN PUD Project—Concept and Organisation*

#### 3.2.1. Goals, Assumptions and Expected Results

The WCN PUD project's main goal was to contribute to the revitalization and reconstruction of the wine cellars of the Negotin municipality by enabling their appropriate use and presentation through urban design. The project aimed to develop urban design strategy and projects for revitalization of four types of WCN in order to help preserving the local tradition of wine production and to support appropriate presentation of this valuable CHS to a broad public. The key focus of the project was the improvement of the WCN's public spaces.

However, acknowledging the importance of having a wider picture of the context, potentials and problems of the Negotin municipality—WCN PUD project aimed not only to physical improvement of WCN and their public spaces, but understood these efforts as steps toward "bringing back the hope to the people of Negotin".

Taking this into account, the WCN PUD project was determined to follow the new inclusive and development oriented CHS paradigm and sustainable tourism approach, and to specifically focus on local peoples' perspectives on WCN in order to provide for sustainable solutions for their development for both wine production and tourism.

We assumed that the understanding urban design as part of WCN sustainable development, and by highlighting the local people's perspective on WCN, will help local communities become a part of revitalization, management and presentation efforts related to WCN. Key questions that WCN PUD sought to answer were:

- How can an urban design process be used to support the involvement of local communities in WCN revitalization and presentation as CHS?
- How can urban design projects be used to present WCN as an existential space of local communities?

Therefore, the aim of the WCN PUD project was to explore the possibilities for appropriate CHS presentation through urban design, and how can local knowledge guide urban design strategy and individual projects, so to reveal and keep local communities' connections with WCN as part of their lives and culture. In order to do so, it was necessary to adopt a participatory approach to urban design for WCN public spaces.

Apart from its main goal, the WCN PUD project had two more ambitions. At the social level, by including participatory perspective and involving local representatives of public and private sectors and local inhabitants in participatory research process for public space design, PUD project aimed to facilitate their better future communication and collaboration as stakeholders in decision making process and management of WCN. At the educational level, being a part of the Master of Architecture Studio course, WCN PUD project aimed to introduce the contemporary tourism and heritage theory and participatory methods and techniques in order to educate students of UBFA as future professionals capable to implement new concepts and methods in CHS management and urban planning and design practice.

### 3.2.2. Theoretical Background and Research Methods

Conceptualization of the WCN PUD project was based on understanding urban design as both process and product of that process, as well as recognizing that it can be conceptualized as a system of decisions in relation to different locations, functions and forms, that all together function in the production of living environment.

Besides that, it was assumed that the knowledge and skills needed for an inclusive and integral approach to CHS presentation should stem from different academic/professional backgrounds to include heritage, tourism, and spatial/landscape planning. Each of them provides specific view on content, use and understanding of heritage as part of wider socio-spatial systems, and contributes to selection of *what* and *why* to present in specific context. In addition, the need to produce design solutions for CHS presentation highlighted the importance of urban design, tourism and heritage management, as disciplines that focus on *how* to present it.

#### Participatory Approach to Urban Design Process

As presented and discussed in the Introduction, the success of local community participation in research, planning or design process depends on many factors. In relation to the specific context and goals of the WCN PUD project it was assumed that for the success of the project it is necessary to enable:

- Inclusion of variety of local stakeholders–actors
- Participation of local people in different phases of project
- Exploration of local people knowledge and relationship with WCN through different participation formats that enable the use of different research techniques

According to this background, the WCN PUD process was organized in four research phases, each having a different role in research and design delivery. These were: Preparation, Field Research, Design, and Presentation and Verification phases, all of which will be presented in more detail in the next section "WCN PUD Process—phases". The participation of local communities was planned in different formats and in all phases of the WCN PUD project (Table 1).

**Table 1.** WCN PUD Research phases, actors, participation formats and local community inputs to design knowledge base.

| PUD Research Phases | Local Community (LC) Involvement—Actors | LC Participation Formats | LC Inputs to PUD Knowledge Base |
|---|---|---|---|
| 1. Preparation | Municipality representatives | Initiation, organization, discussion | Documentation: studies, plans, strategies |
| 2. Field research | Municipality level public, private, civic sector representatives Local level: commissioners, inhabitants | Public meetings and gatherings, guided torus by local commissioners, direct contact with locals: Data collection: survey, questionnaire, semi-structured interviews, in-situ guided tours | General: Social, economic and spatial problems and potentials at municipality and local level Specific: Stories and Places-information and values of specific WCN locations: potential resources and solutions for presentation |
| 3. Design | Municipality level public, private, civic sector representatives Local level: commissioners, inhabitants | Direct contact with local inhabitants Meeting with Municipality representatives | Additional specific information on locations Critical review of draft WCN PUD strategies and projects |
| 4. Presentation and verification | Municipality level public, private, civic sector representatives Local level: commissioners, inhabitants | Public exhibitions and discussion in Negotin, Rajac, Rogljevo, Stubik, Smedovac.. Survey | Critical review of draft WCN PUD strategies and projects |

The purpose of the *Preparation* phase was to set the basis for the project in theoretical, methodological and organizational terms. In this phase participation of the representatives from the Municipality of Negotin was crucial in providing necessary strategic and planning documents and indicating already recognized development and organizational problems.

The organized *Field research* phase was supposed to take place in the city of Negotin and in four selected locations WCN involving local stakeholders from public, private and civic sector. It was planned in order to provide collection of data through various research techniques. In order to provide for the rich data base for understanding of local peoples' perspectives on WCN as the knowledge base for urban design, the research team conducted: questionnaires, semi-structured and unstructured interviews, walking tours, as well as photo and spatial mapping as research techniques.

The *Design* phase of the project was based on the conclusions of the desk and field research, and included development of the urban design strategies and related projects for all four WCN locations. During this phase, communication with individual WCN local communities was planned to occur on an informal basis, conducted by each research group, following the dynamic of their project development. Besides that, Christian Norbert Shultz's theory of existential space was adopted as a tool to overcome limits of direct involvement of local communities in the design process, and to appropriately translate their complex relationships with WCN into design proposals.

In order to verify if the research team managed to effectively integrate local perceptions and values on WCN into urban design projects, the *Presentation and Verification* phase was planned. It was important to enable local communities to participate in the evaluation of the design proposals for their WCN. This was done through the organization of formal exhibitions in all WCN locations as well as in the city of Negotin. These exhibitions were used as the opportunity for local people to evaluate design projects through questionnaires, and to gain their opinion on how to improve design proposals.

Existential Space Perspective to Urban Design Projects

The WCN PUD project adopted Christian Norbert Shultz's theory of existential space as the basis for translating the local people's relationship with WCN space. This theory was considered appropriate since it relates people to their environment through understanding of architectural space as a support and a reflection of human existence.

Schluz [71] defines existential space as a relatively stable system of perceptual schemata—image of the environment. He argues that people exist spatially, and that existential space represents human existence in the world. He builds upon the work of

Piage and his notion that our "space consciousness" is based on "operative schemata" as our experiences of the artefacts in the environment. These "schemata" are made up of elements, which possess certain invariance, and are socially or culturally conditioned. All this together makes up a man's "image" of his environment, as a stable system of three-dimensional relations between objects of various meanings.

The structure of human existence contains two aspects, one of which is "abstract" and the other "concrete". The abstract aspect consists of general schemes of the topological or geometric type. The specific aspect, in turn, refers to the notion of "elements of the environment" such as landscape, urban forms, buildings and physical objects. The theory of existential space contains both of these aspects and defines the architectural space as the concretization of existential space.

Schultz interprets basic assumptions of the psychology of perception suggesting that elementary organizational "schemata" consist of central places (proximity), directions or paths (continuity) and areas or domains (enclosure) and relate them to architectural space. People orient themselves based on an understanding of these topological relations. Centers, paths and domains represent the basic schemes of orientation, which means that they are constituent parts that make up the existential space, and architectural space as its concretization. Centers/Places are "goals or foci where we experience the meaningful events of our existence, but they are also points of departure from which we orient ourselves ad take possession of the environment" [71]. Paths can have many forms and directions and divide the environment into domains. The characteristic of each path is its continuity. The domains are potential places of man's activities and have certain unifying function. When centers, paths and domains combine, space becomes a real dimension of human existence and can materialize in a form of architectural space. Elements of existential space appear on several levels within a hierarchy that has geography and landscape on one and smaller objects on the opposite side. For the architectural space as a concretization of the existential space, relevant are landscape, settlement and building levels.

These basic concepts, elements and levels of existential space were further used for interpretation and integration of identified local communities' relations to space in order to produce urban design proposals that reflect their ways of existence in specific local. Important implication that came from this point of view was to approach WCN as part of the rural—cultural landscape and analyze it in relation to related village and other relevant elements of the territory.

### 3.2.3. WCN PUD Project Organization
WCN PUD Project Team

The WCN PUD project has been conducted as a joint effort of academic research team and a team consisting of representatives from Negotin municipality. The academic research team of WCN PUD project consisted of two senior researchers/mentors and professors from UBFA and University of Sasari Faculty of Architecture from Aghero, four researchers/tutors from UBFA and a group of 16 junior researchers/master level students from UBFA. Junior researchers were organized into 4 research groups, each focused on research and design for specific WCN location, while senior researchers and tutors worked with and coordinated all four research and design groups. The Negotin municipality team included representatives from the Municipality of Negotin as well as local commissioners for each location.

In parallel, the research team from The Faculty of Forestry, Department of Landscape Architecture and Horticulture were conducting the educational project—"Study of the Cultural Landscape of the Wine Cellars region". The representatives of this research team joined the WCN PUD team in one of the field works, and also collaborated in research process thorough discussions on specific issues. These joint efforts contributed to the interdisciplinary character and strengthened landscape perspective of the WCN PUD project.

Besides that, discussions on how to approach specific issues in WCNPUD project were conducted with a group of external international experts in order to clarify and test some of the proposals in relation to their theoretical and professional expertise.

Limiting Factors in WCN PUD Process Organization

There were three limiting factors that were taken into account in conceptualizing WCN PUD process, and these are: time, distance + financing, and organizational constraints. The time limitation is related to the fact that WCN PUD is a UBFA educational project that was supposed start in February and finish in May. Therefore, all phases of the project were supposed to be delivered within this time-limit.

Another limiting factor was the distance: Municipality of Negotin is 300 km away from the city of Belgrade. Besides that, to organize visits for the whole research team was also an important financial issue. Therefore, only two planned and financially supported visits to a location were possible to plan in advance. Privately organized visits to locations were considered as a possibility.

In relation to organization, two limits were the most important. One considers difficulties to include representatives from National Heritage Protection institute in the process. For this reason, knowledge-based design approach and the precautionary principle in relation to treatment of buildings and structures that form the CHS have been applied. This has led to detailed document analysis of public documents and expert studies in order to set the limits of all urban design the interventions.

The other problem was related to the limited possibilities to inform and include wither local community in PUD process. A research team from Belgrade couldn't rely on ICT techniques, because a very small number of inhabitants are using the internet. So we had to rely on our local partners from the Municipality of Negotin to do this work. Key role in overcoming this problem had the local commissioners for each location. They helped in organizing communication with local communities, and also functioned as key informants and guides to the research team.

### 3.3. WCN PUD Process—Phases

3.3.1. Preparation

After the initiation and formalization of the WCN project in May 2014, the first exploratory study trip to Negotin was organized in July 2014. The interdisciplinary team of professors and experts from Serbia (UBFA) and from Italy (Italian embassy, Faculty of Architecture, University of Sassari and from the Faculty of Architecture, Roma Tre University), as a know-how partner in the project, have visited the variety of WCN locations, and had meetings with the representatives from the Municipality of Negotin. This exploratory study trip helped selecting specific locations of WCN to be further examined through WCN PUD project.

During October, November and December 2014 preparation activities continued in the form of desk-research and consultations between partners in order to identify organizational potentials and constraints. Desk research included: (a) The review and systematization of the WCN studies and planning documentation (provided by the Municipality of Negotin) in order to identify WCN CHS values to be protected as a framework for design interventions, and (b) The critical review of the existing scientific research on WCN in order to identify research gaps and clarify the approach to research for design. Based on all information gathered and discussed by partners, the research and organization strategy of the project has been designed.

In addition, this preparation phase included education of junior researchers (students) for conducting participative research. For that reason, besides lectures on the theoretical basis for community participation in the development process, two workshops with experts in the field have been organized. The first one, "participative urban design", was organized in order identify relevant stakeholders, to select appropriate research techniques and plan for the data collection. The second workshop "Participation in urban development

planning" was focused on educating junior researchers how to work with local communities and implement different research techniques in the specific context.

In this phase, the research team designed questionnaires, selected key guiding questions for the interviews, planned the field work and developed the case study protocol in order enable collection of comparable data/information on WCN at different locations. Since four different locations were selected for detailed examination, four research groups were formed in order to collect data.

### 3.3.2. Field Research

The site visit and the data collection and organization took place between 6–8th of March 2015. The UBFA research team, visited the city of Negotin and all four selected WCN locations, and performed different research techniques in order to collect diverse information on WCN space and local people's relation to this space. In this way wider urban design knowledge base was formed (Figure 3).

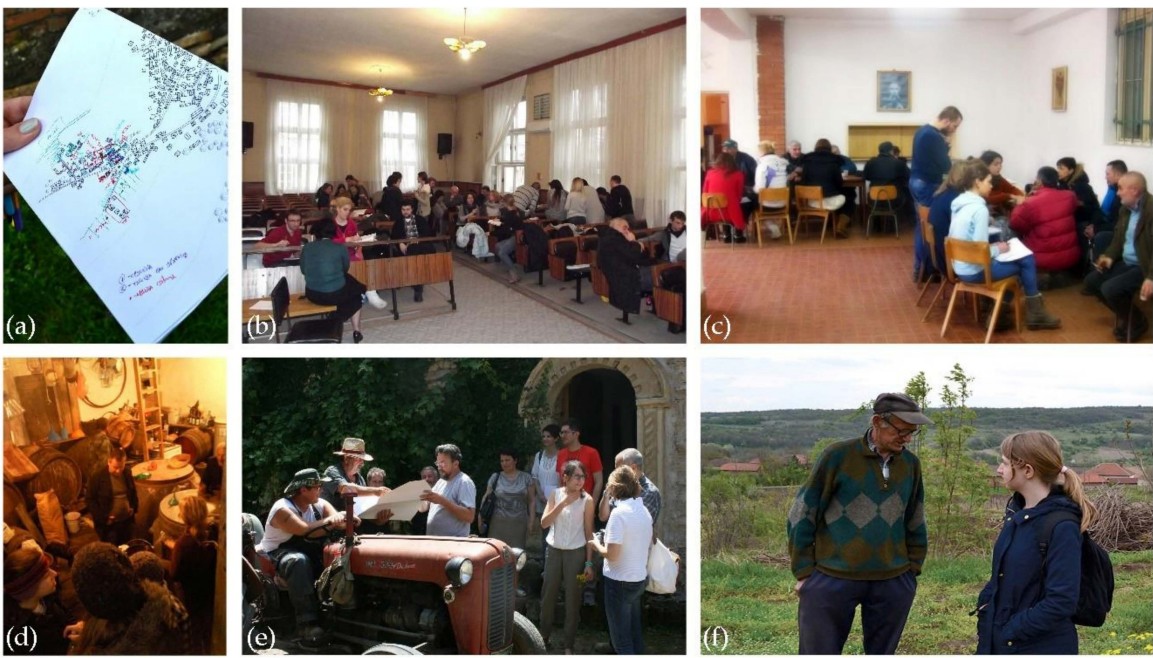

**Figure 3.** WCN PUD Data collection and community involvement: (**a**) Field work: mapping and observation; (**b**) Public meeting and survey with stakeholders in Negotin; (**c**) Organized interviews in elementary school in Rajac; (**d**) Organized interviews with wine producers in wine-cellar in Rogljevo; (**e**) Guided tour in Rogljevo; (**f**) On-site interviews with inhabitants in Smedovac.

The official presentation of the WCN PUD project and the public meeting with stakeholders from public, private and civic sector was organized in the city of Negotin, as a central city of the Municipality. This meeting was used for conducting the survey on stakeholders' opinions on WCN, their meaning, importance, quality, relationship with the villages, and recommendation for improvement. Besides that, all present stakeholders were asked to identify key potentials and problems for WCN development. Based on this it was further possible to analyze similarities and differences in opinion of different stakeholders, mainly those from the Municipality level.

The similar research further took place in villages of Rogljevo and Rajac. In order to directly involve people who live in these villages in the research process, local commissioners organized meetings that were used for more detailed data collection. In Rajac, the meeting with local community was organized in the local school, while in Rogljevo, the meeting with local producers and inhabitants was organized in one of the active wine-cellars. Besides conducting survey and information on key potentials and problems, these

meetings additionally enabled researchers to conduct semi-structured interviews in order to get information that will help them reveal historical and contemporary relations with space and how do local people perceive, use and feel about WCN. They were able to hear stories and conduct mapping of places that were special for locals for different reasons.

Besides that, in both villages Rogljevo and Rajac the guided tours around WCNs were organized by local commissioners and used to better explain their personal and community relationship with place. In the villages of Štubik and Smedovac, the research was not conducted through organized public gathering. The research groups made personal contacts with local inhabitants and conducted surveys and interviews.

In all WCN locations research groups collected video and photo documentation and conducted mapping of existing structures, their use and level of activity. This was especially important for villages of Štubik and Smedovac, where previous mappings have never been done before. Since this mapping followed the methodology of the National Cultural Heritage Institute developed for valorisation of Rajac and Rogljevo WCNs, PUD project made a significant contribution to further conservation and adaptive use of WCN in the other two villages.

The information was collected from 50 members of the local community: 13 representatives of the Municipality of Negotin 8 wine producers (private sector) and 29 local inhabitants were interviewed [57].

The results of the survey showed that [57]:

- Both cultural and natural values of the region are important to participants
- Almost all participants had positive attitude towards the development of WCN for tourism
- Great majority of participants perceived WCN as important but recognized the need for their improvement in terms of renovation, public infrastructure and safety.
- Opinion about the character of the future relationship between WCN and village varied in accordance with the specificity of each case. It was highly rated in Rajac, Rogljevo and Smedovac, but inhabitants of Štubik didn't show any interest in WCN.

The results of the questionnaire about the hierarchy of potentials and problems for sustainable development of WCN [57] enabled identification of key problems and potentials that were of major importance for all stakeholder groups, such as: Problems of depopulation and bad infrastructure, lack of gathering places; Potentials: Local communities' pride of having important CHS on the territory; Tradition in wine production; Positive attitude towards tourism; organization of traditional events.

Besides that, they revealed the differences between groups of stakeholders in relation to their perception of priority problems to be addressed in WCN development. For the public sector the key problems were related to the funding of WCN reconstruction, lack of qualified personnel, and problematic relationship with both private sector and the Department for Heritage protection. For the private sector (wine producers and tourism industry) the key problems were poor promotion and support for wine production and tourism; while the local communities identified as key problems those related to the quality of life, such as lack of public activities and social services, and a low quality of public space maintenance.

### 3.3.3. Design

Based on the collected information and results of the survey and hierarchy questionnaire, as well as the results of the in-situ research with local communities, each research group proposed the draft urban design strategies for specific WCN. In doing so they took into account potentials and limits to their development as well as the specificities of how local people relate to space. These spatial strategies were further operationalized through specific public space design projects for the locations that were identified through the research as an important expression of local people's existential space.

During this period, research groups made individual trips to researched WCN in order to collect more specific information and test their ideas on public space design with

local inhabitants. Besides that, senior researchers had a meeting with the municipality officials and discussed the quality of the design proposals in relation to the expressed local knowledge and attitudes. All of this enabled research groups to make corrections in order to better harmonize different interests in space and strengthen the authenticity of the WCN presentation as both CHS and the existential space of local communities.

### 3.3.4. Presentation and Verification

Presentation and verification phase took place between 21–23rd of May 2015. Research groups presented their research and urban design proposals in the city of Negotin as well as in Rajac, Rogljevo, Štubik and Smedovac villages. These exhibitions were used for verification of the projects by local people. Verification has been conducted through a survey among the inhabitants that were previously included in the field research phase. They expressed high opinion on how their perception and attitude on WCN has been integrated into Urban design strategy and individual projects. Official verification of the project results was also expressed through the Municipality of Negotin initiative for the realization of one of the proposed projects (Figure 4).

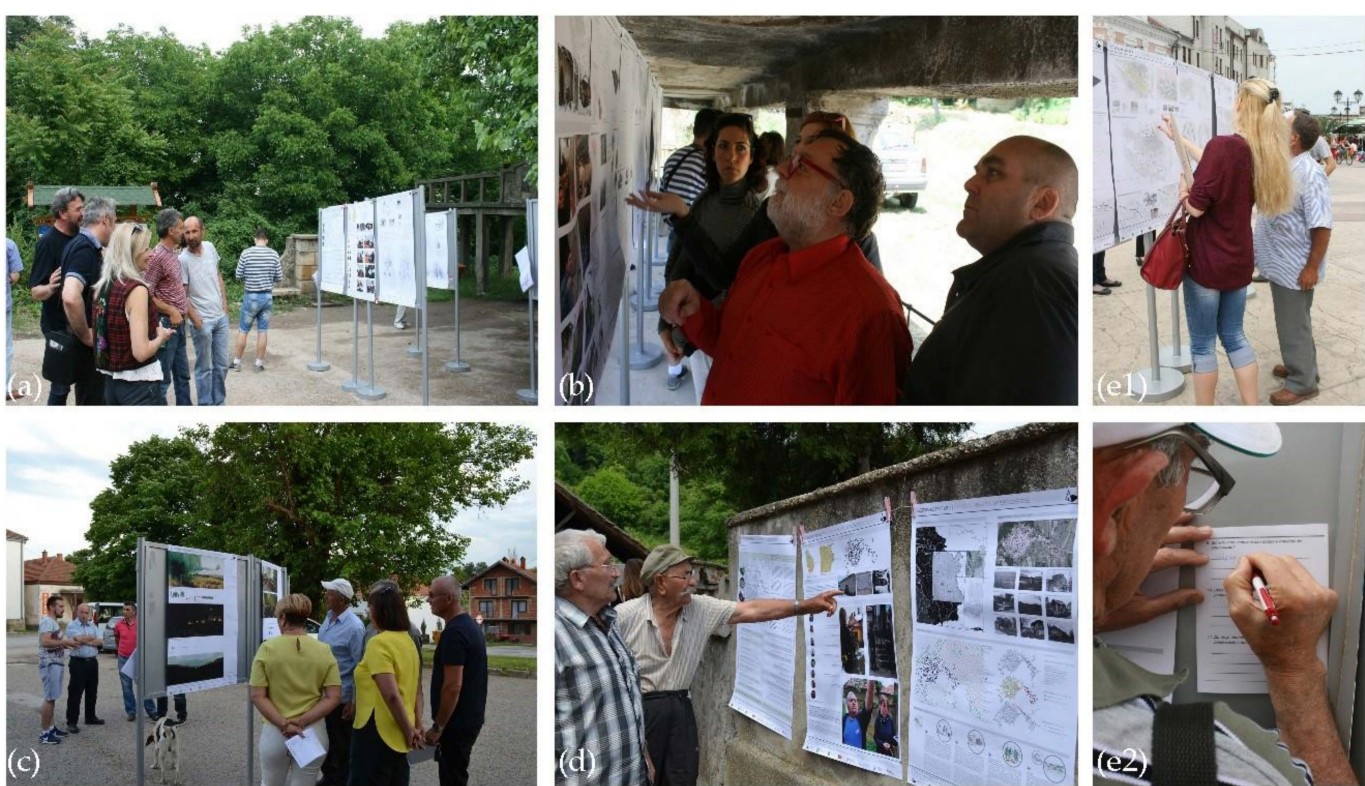

**Figure 4.** WCN PUD presentation and verification: (**a**) Rajac; (**b**) Rogljevo; (**c**) Stubik; (**d**) Smedovac; (**e1,e2**) Negotin.

Besides municipality of Negotin and local communities' verification, the WCN PUD project has been recognized as important for advancing the planning, research and profession in Serbia, and awarded with the 1st prize at the 28th International Planning Exhibition in Niš.

### 3.4. WCN PUD Projects

In this section WCN PUD projects from each location will be presented in relation to context, key inputs to the knowledge base, UD strategy and design projects as well as related forms of presentation.

### 3.4.1. Rajac

Context

Rajac is a village located 22 km south of Negotin, on the left valley side of the Timok river. It is an agricultural village with a long tradition in wine production. There are 194 households in the settlement with 436 inhabitants (census 2011), facing a negative population trend. The median age of the population is 56.5 years.

Rajac wine cellars (Rajačke pivnice) are located 2 km to the west of the village Rajac, on the hill, and close to the vineyards. This temporary, compound settlement was built in the 18th and 19th century and is in active use since then. It was built spontaneously around the central square with a fountain, following the terrain. Buildings are often clustered in a row or in blocks, both the ground floor ones and the stories ones. The cellars are mostly built out of the local limestone and are covered with tiles. The WC of Rajac consist of 169 buildings with 166 cellars and 3 distilleries, out of which 57 cellars are still used for making wine, 16 are in ruins, while 19 are used for tourism purposes as restaurants, taverns, inns, exhibitions, etc. In the vicinity of the complex of Rajac wine cellars, the village cemetery Beli Breg is located. Dating from the 17th century, it has been recognized and protected as cultural value for its 200 old decorated tombstones [51]. The Rajac wine cellars and cemetery were declared cultural heritage in 1980, classified as an area cultural-historic ensemble of outstanding value in the Republic of Serbia, and included in UNESCO Tentative List in 2010 [51] (Figure 5).

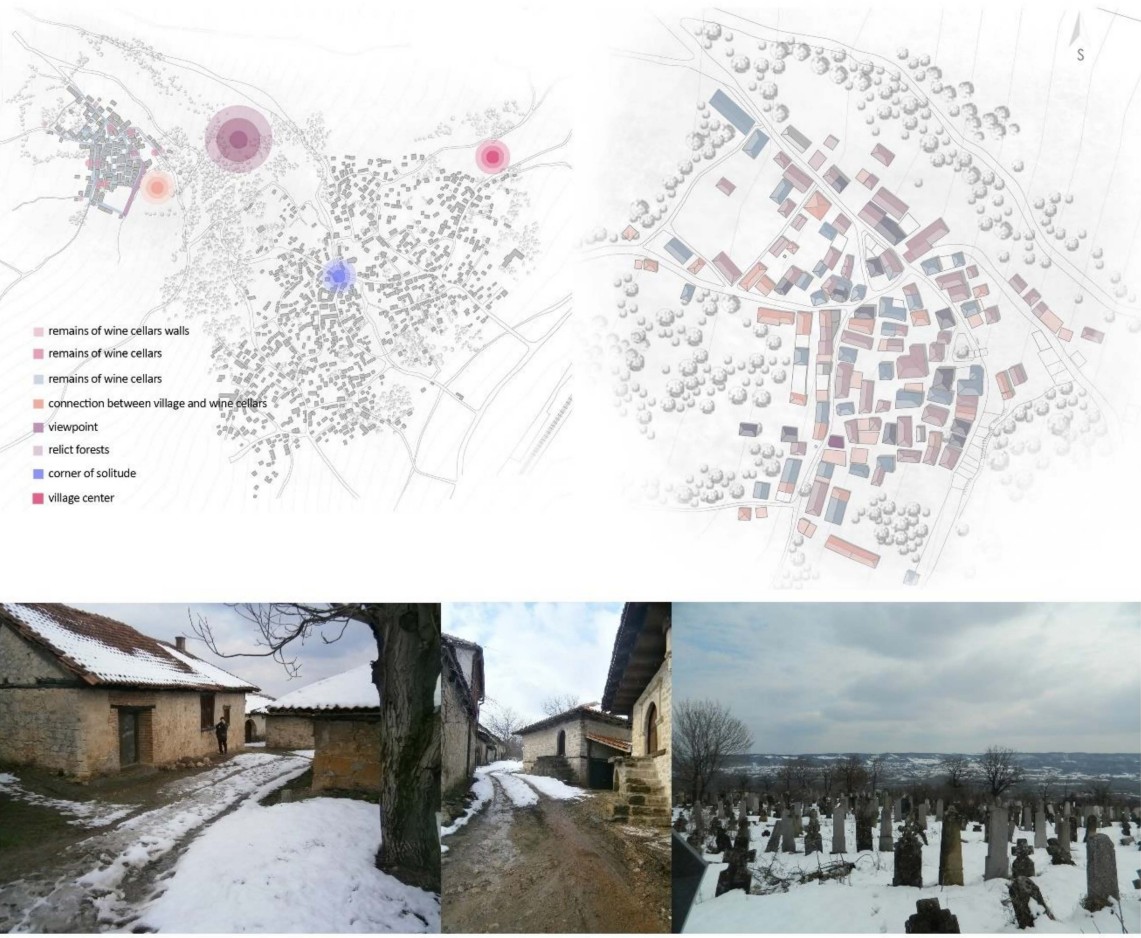

**Figure 5.** Rajac—Wine-cellars location, research area and community inputs.

Project

- Local communities' key inputs to WCN presentation through UD

Inputs were gained from both municipality and local-village participants in the research process. Different forms of the participation of local community revealed: *(1) Key problems*: lack of infrastructure and gathering places as important for both quality of life and tourism; *(2) Key potentials*: perception of WC as important for their identity, everyday life and future wellbeing; wine production as economic and cultural tradition; local initiatives in tourism; protected CHS; *(3) Stories to be told:* (a) WC as places of celebrating and mourning in life of locals; (b) Stories about variety of links between the village and the WC, (c) Popov kladenac; *(4) Places to be revealed*: (a) Local materials as resources to be used for pavement, (b) Night ambiance of WCN to be expressed through lighting, (c) Hidden paths between the village and the WCN to be revealed and marked

- UD Spatial Strategy (Figure 6)

The spatial strategy for WC was based on the recognition that: (a) Conservation plan for WCN exists and provides general guidelines for design; (b) Local community expressed the interest and have the capacity to be involved in the management and presentation of WCN—making living heritage concept possible, (c) Local inhabitants identified important places and hidden links at the different existential levels as important for future use, development and presentation of WCN. Therefore, the focus of the spatial strategy for presentation of WCN was to RE-CONNECT WC at different levels: with other elements of rural landscape, with village of Rajac, as well as to establish better links within the area of WC.

- UD Projects for WCN Presentation (Figure 6):
  1. "Belvedere"—scenic viewpoint: designed for presenting the WC as a part of the rural landscape, with the use of local materials for ambient and educational purpose.
  2. "Gathering place": designed as a "living display" of WC and a starting point for formal walking and guided tours. It is also supposed to function as place for lectures, events and activities for visitors. The place is equipped with new street lighting and furniture. Complex pattern of pavement with local materials was used for ambient and educational purpose.
  3. "Popov kladenac" (the priest's well): Designed to support guided storytelling/lectures as well as to enable introspection, solitude and atmospheric experience
  4. "Places in-between": designed to support everyday life and learning activities, as well as informal "display" of local life.
  5. "Path from the Village to Wine cellars": designed to support walking and guided tours as well as multisensory experience of everyday local communication between two important areas of Rajac. It connects important focal points and is equipped with panels and signposts. Since visitor center is planned in the village—carriage can be used by tourists to get to wine cellars by using this route. Path is equipped with guideposts, benches and places to rest that can be used as guided tour stops.
  6. "Paths to woods": these paths are designed by use of different local materials (gravel, grass, wood) and equipped by panels in order to present local nature and relationships within the cultural landscape. They connect specific places identified by locals (viewpoints, the murmur of water, bird habitats).
  7. "Paths through Wine cellars": designed in order to present WC settlement as "living display" of wine-production and wine culture and to support a variety of touristic activities. New lighting was introduced, and local materials were used for ambient and educational purpose.

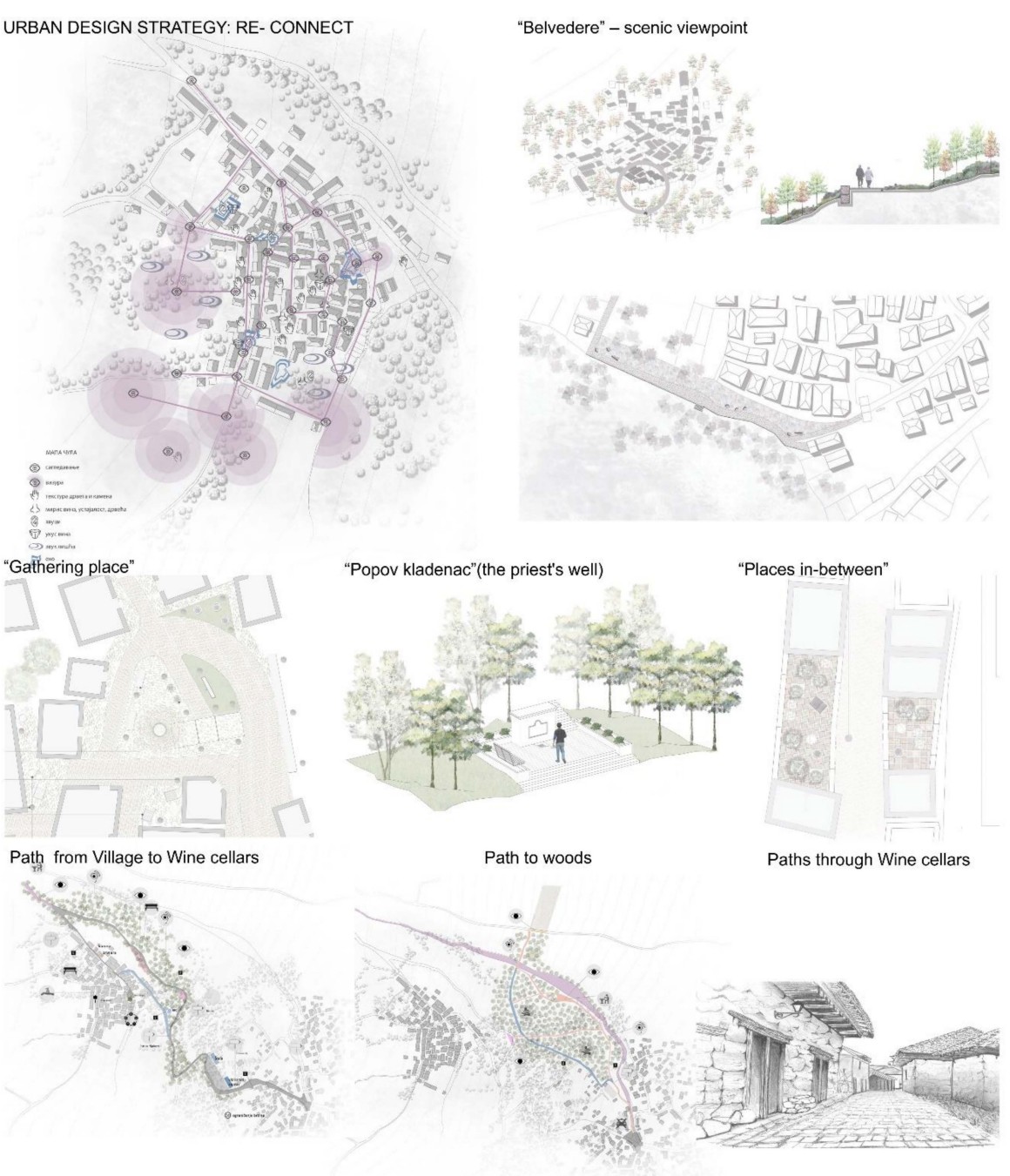

**Figure 6.** Rajac–Urban design strategy and projects for WCN presentation.

Integration of Local Knowledge into UD

Different forms of participation of local community resulted in the formation of complex knowledge base for urban design and WC presentation. Urban design projects were conceptualized as the expression of local people's perception of WC as part of their existential space. The analysis of the locations identified by locals shows that they perceive WC as an integral part of the wider whole- their existential space includes the village, WC and rural landscape of Rajac village, and it was important to present it as such. Types of existential space that locals identified as important to be presented are also complex and include centers, paths and areas in complex relations.

### 3.4.2. Rogljevo

Context

Rogljevo is an agricultural village located 14 km south of Negotin, on the left valley side of the Timok river. It is one of the oldest villages in the region, with a long tradition in wine production. In 1863, there were 600 people living in Rogljevo, and the vineyards were spread over more than 100 hectares, with annual production of nearly 400 thousand litres. Today, there are 45 households in the settlement with 123 inhabitants (census 2011), with decreasing population trend. The median age of the population is 55.3 years.

Rogljevo wine cellars (Rogljevačke pivnice) are one of the most famous and best-preserved of its kind in the region. They were built in the 19th century, close to the main village of Rogljevo, on a nearby hill with vineyards. Settlement is characteristic for its irregular streets, and a center that is densely built around the sacred tree and the common table, the folk table and the well. The compound is composed out of around 122 buildings, built mainly out of limestone. The majority of cellars, 40 of them, were built between 1859 and 1890. Those are mostly large structures of the vast rectangular ground plan and of substantial height, with simply decorated but imposing arched portals. In this compound, a great number of cellars are still in their original use for wine making and storage. Several wine cellars (8) are recently converted into restaurants, taverns, exhibition or accommodation places, and used for tourism. The residents and owners of the wine cellars are very interested in the active promotion of their viticultural heritage. Rogljevske Pivnice were declared cultural heritage in 1983, classified as an area cultural-historic ensemble of outstanding value in the Republic of Serbia, and included in UNESCO Tentative List in 2010 [51] (Figure 7).

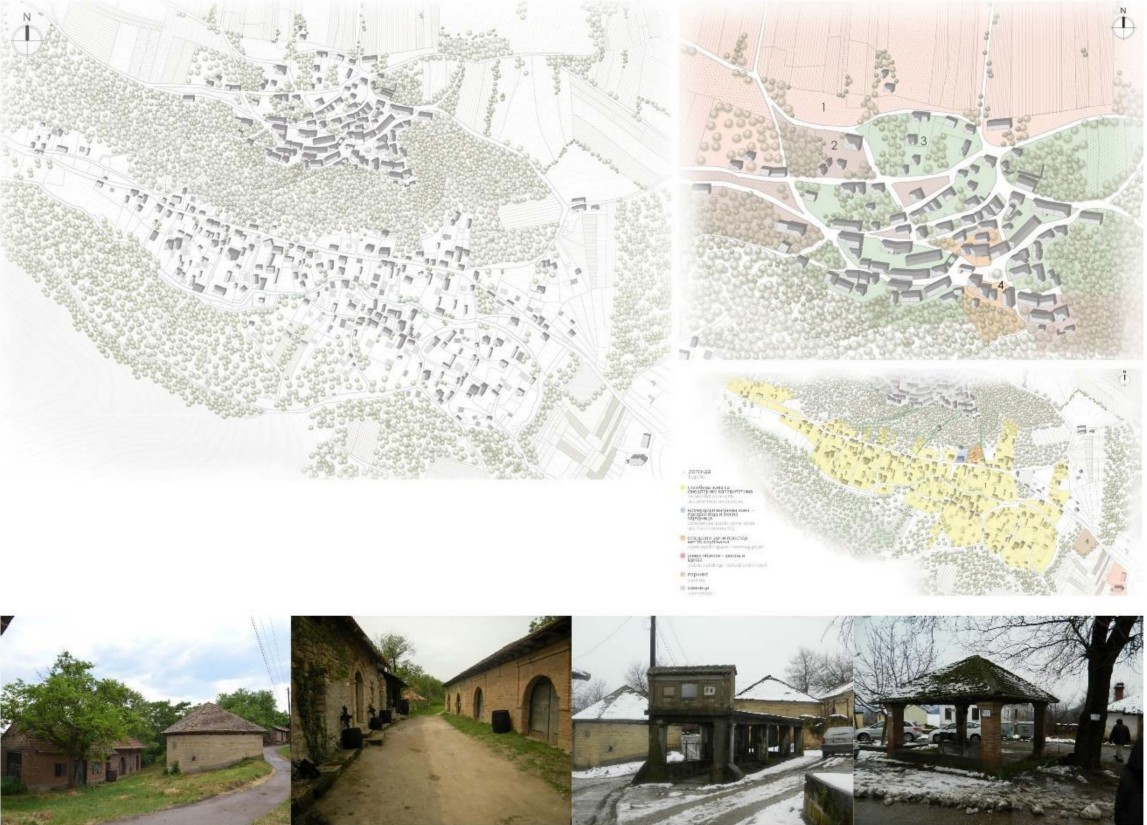

**Figure 7.** Rogljevo–Wine-cellars location, research area and community inputs.

Project

- Local communities' key inputs to WCN presentation through UD

Inputs were gained from both municipality and local-village participants in the research process. Different forms of the participation of local community revealed: *(1) Key problems*: lack of infrastructure and gathering places as important for both quality of life and tourism; *(2) Key potentials*: perception of WC as important for their identity, everyday life and future wellbeing; wine production as economic and cultural tradition; local initiatives in tourism; protected CHS; *(3) Stories to be told:* (a) PEOPLE and WINE—People need sounds—wine needs silence; (4) *Places to be revealed*: (a) Local materials as resources to be used for pavement, (b) Night ambiance of WCN to be expressed through lighting.

- UD Spatial Strategy (Figure 8)

The spatial strategy for WC was based on the recognition that: (a) Conservation plan for WC exists and provides general guidelines for design; (b) Local community expressed the interest and have the capacity to be involved in the management and presentation of WCN—making living heritage concept possible, (c) Local inhabitants identified important places at different existential levels as important for improving future use and presentation of WCN. The focus of the spatial strategy for presentation of WCN followed existing activities, perceptions and stories of locals, and aimed to present the balance between *sound and silence* in the life of people and wine.

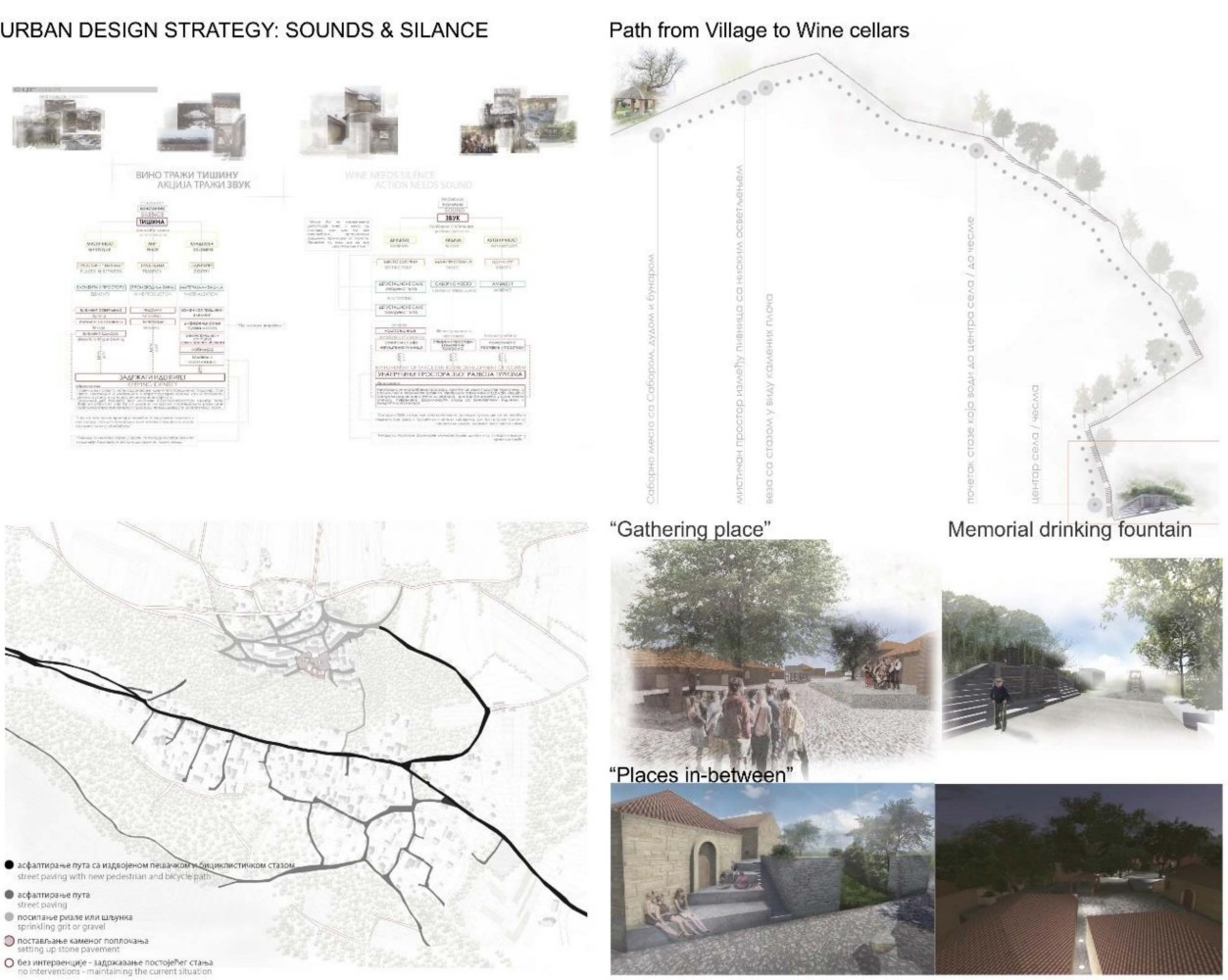

**Figure 8.** Rogljevo–Urban design strategy and projects for WCN presentation.

- UD Projects for WCN presentation (Figure 8)
    1. "Path from the Village to Wine cellars"—presentation of local peoples' everyday life and their functional and symbolical relation with the wine cellars is enabled by design of the path between "sounds and silence". This path passes through different natural and built structures in WC and surroundings in which each stop "tells the story", supported with specific design elements.
    2. "Gathering place"—this place is the "heart" of the WC, where local people gather during religious holidays and for all local events and celebrations. Several important structures are located here (Sabor, well and mulberry tree) and they form the specific ambient together with surrounding wine cellars. Design of pavement, street- light, and benches aims to present and shed light on both tangible and intangible values of this place as a "living display". Also, it aims to enable a variety of touristic and presentational activities to take place (lectures, workshops, degustation and displays of local products), while at the same time keeping alive local traditional events and gatherings. This is a symbolic place of sounds in WC settlement of silence.
    3. Memorial drinking fountain—this is an important and symbolic place for local community and design project aimed to reveal it as such. Use of the greenery and local materials for walls and pavement, as well as discrete street light enable protection of the authenticity of place while at the same time enabling new activities of gathering in silence—for memory, meditation and contact with nature. At the same time this place is used as the starting point for tourist route that links village and WC.
    4. "Places in-between"—strategic design, lighting and the use of narrow segments between WC, amplifies the mysterious atmosphere and aims to support the creation of memorable touristic experiences.

Integration of Local Knowledge into UD

Character and level of participation of local community from Rogljevo were similar to Rajac and reflected their interest and enthusiasm for future WC development as both wine-production and touristic place. This resulted in the formation of complex knowledge base for urban design and WC presentation. It was possible to conceptualize urban design projects as the expression of local people's perception of WC as part of their existential space. The similar distance between village and WC resulted in the fact that local people perceive WC as an integral part of the wider whole—their existential space includes the village, WC and rural landscape with vineyards, and it was important to present it as such. Again, types of existential space, which locals identified as important to be presented, included centers, paths and areas in their complex relations.

### 3.4.3. Štubik
Context

Štubik is a big village located 22 km north-west of Negotin, famous for farming, cattle breeding and wine production. There are 317 households in the settlement with 825 inhabitants (census 2011), facing a negative population trend. The median age of the population is 47.9 years.

The wine cellars of the village of Stubik are located about 5 km from the city of Negotin, and about 15 km from the village of the same name. There used to be about 400 buildings, to be reduced to 260 in the middle of the 20th century, and only 39 have been preserved to this day. Today, the regional road divides the settlement spatially into two parts and there is no activity in the wine cellars that remained. Built until the first decade of the 20th century, the wine cellars of Stubik are of the oldest type preserved. Unlike the representative wine cellars of Rajac and Rogljevo, that were made of brick, the wine cellars of Stubik were built as a ground floor, wooden buildings with a porch, and two-story buildings with doxa. In the latter case, the buildings had the basements built of crushed

stone, the two-story residential part built of the rammed earth, and they were covered with a tiled roof. The Štubičke Pivnice were declared cultural heritage in 1980, classified as an area cultural-historic ensemble of outstanding value in the Republic of Serbia and included in UNESCO Tentative List in 2010 [51] (Figure 9).

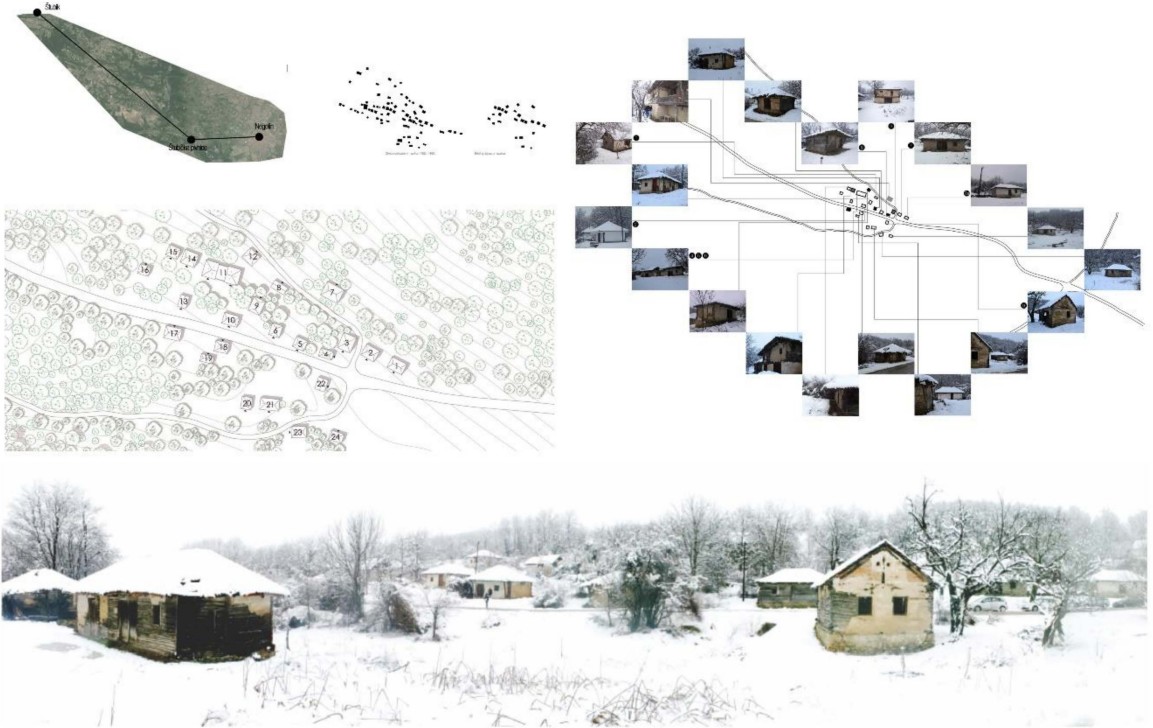

**Figure 9.** Stubik–Wine-cellars location, research area and community inputs.

Today, despite their neglect and degradation Stubicke pivnice retained significant architectural and artistic values. Unfortunately, the conservation activities have not yet been applied, and their survival is endangered since they are not used and were left to decline.

Project

- Local communities' key inputs to WCN presentation through UD

Unlike for other WCN, local communities inputs were gained only from the municipality level stakeholders, namely from the public sector. They recognized the importance of Štubik wine cellars as important CHS, part of local identity, and potential for tourism development, and envision them as a key future touristic centre for presentation of wine-production of Negotin municipality. This differs from the Republic Institut for the cultural heritage's vision of presenting WC of Štubik as "etno village". Through questionnaires with stakeholders, key problems and potentials have been revealed: *(1) Key problems:* lack of infrastructure and protection of WC; neglect and disconnection with village; *(2) Key potentials:* perception of WC as important for the Negotin municipality identity, protected CHS.

- UD Spatial Strategy (Figure 10)

Although designated as a cultural heritage site, and being part of WCNs proposed for UNESCO WHS, the conservation plan for WC of Štubik does not exist to provide guidelines for design. Besides that, local people didn't express any interest in their future use and development. Big distance between village and WC and socio-economic situation—dissociated local community of Štubik from their WC. It was not realistic to expect their active engagement in future development and presentation of WC for tourism, although their inclusion as stakeholders in development process remains important. Besides that,

the uncertainty of the form and purpose of future development was present, since conflict visions of future development of WC exist. In such context, besides conducting the research study on WC buildings and spaces as a basis for design, UD spatial strategy, called "IN-BETWEEN" aimed to activate interstitial spaces in order to support informal and temporary use of WC for touristic presentation and education.

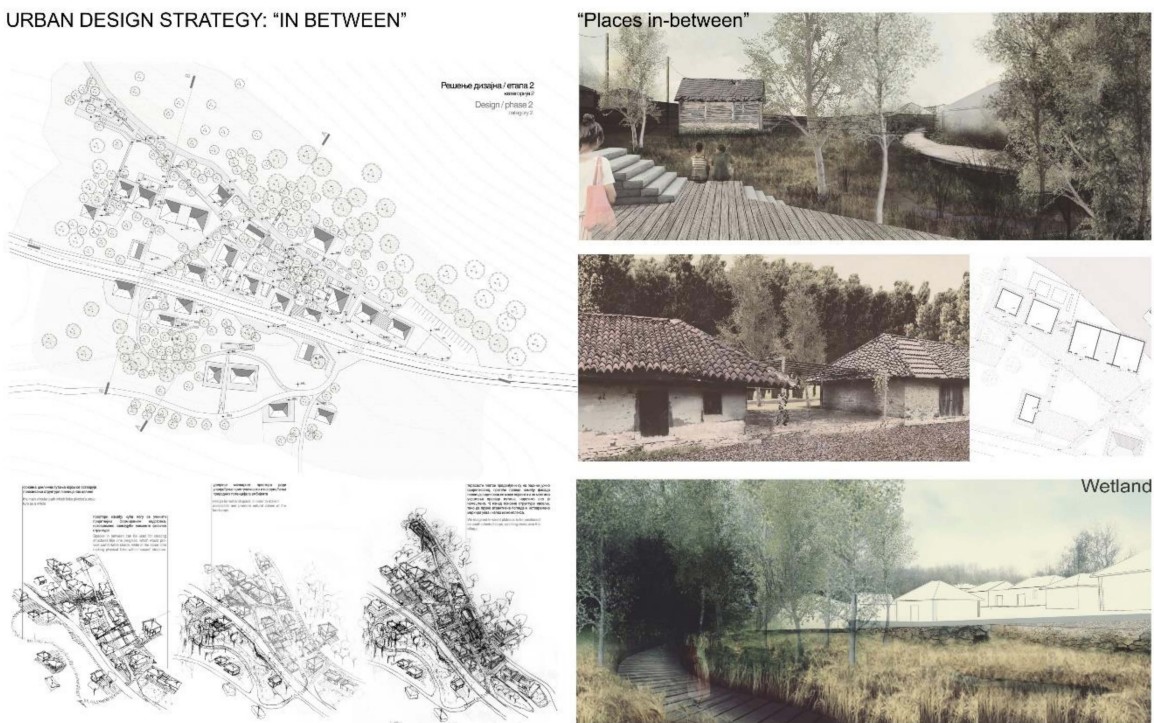

**Figure 10.** Stubik–Urban design strategy and projects for WCN presentation.

- UD Projects for WCN presentation (Figure 10)
    1. "Places in-between"—design of the canopy and pavement aimed to articulate space for lectures, events, activities for visitors as forms of CHS presentation.
    2. "Wetland"- design of the wooden path which connects segments of WC aimed to present specific relation between nature and local culture of wine production, and to create memorable experiences in walking and guided tours by contrasting ground levels, materials, vision and sounds of present and past.

Integration of Local Knowledge into UD

Integration of local knowledge relates only to public sector representatives from Negotin and the absence of participation of inhabitants of Štubik village implied the formation of very basic knowledge base for urban design and WC presentation. Therefore, it was not possible to conceptualized urban design projects as the expression of local people's perception of WC as part of their existential space.

3.4.4. Smedovac
Context

Smedovac is located near the villages of Rajac and Rogljevo, about 25 km from Negotin. It is one of the smallest villages in the region, with a population of 112 inhabitants. It also has a very old population, with the median age of the population of 60. Although, as other villages in the region, famous for its viticulture and wine production in the past, nowadays this activity is abandoned.

The wine cellars in Smedovac (Smedovačke pivnice) are unique because they were built within the village, unlike the other wine cellars in the region. They are located at the

very entrance to the village, on the road between Rogljevo and Rajac. The WC complex is a structural part of the village, and the border between them is not clearly defined. A large number of wine cellars are abandoned, and a considerable number is devastated. Architecture of the wine cellars of Smedovac is more diverse than in other compounds—they are built of timber, sandstone, brick, and some of them are plastered, painted and decorated. However, the village and wine cellars of Smedovac are not perceived as important for tourism development and revitalization in the official Municipal plans and strategies. This is likely due to a negative demographic trends and population age (Figure 11).

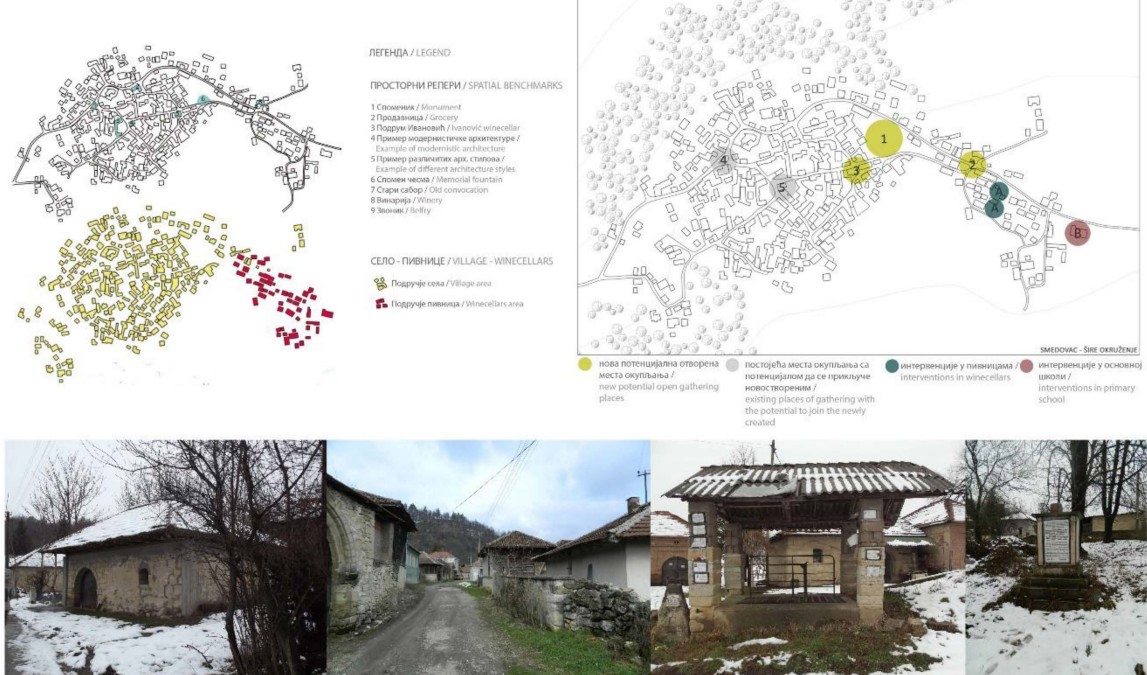

**Figure 11.** Smedovac–Wine-cellars location, research area and community inputs.

Project

- Local communities' key inputs to WCN presentation through UD

Specific inputs for the further development of WC were gained mainly from local-village participants in the research process. Municipality level participants recognized the importance but do not conceive Smedovac and its WC as the key priority for tourism development. Participation of local community revealed: *(1) Key problems:* older population that is not able to continue with wine production; lack of infrastructure, services and gathering places as important for the quality of life and attraction of young people; *(2) Key potentials:* perception of WC as important for their identity, everyday life and future wellbeing; wine production as economic and cultural tradition; enthusiasm of local people; location of WC as the extension of the village; *(3) Stories to be told*: (a) memories of village when it was large, with many young people and school; (b) memories of wine production, (c) memories of historic events; *(4) Places to be revealed:* (a) School, (b) wine street, (c) drinking fountain (d) memorial place, (e) local materials as resources for design, (f) night ambiance of WCN to be expressed through lighting.

- UD Spatial Strategy (Figure 12)

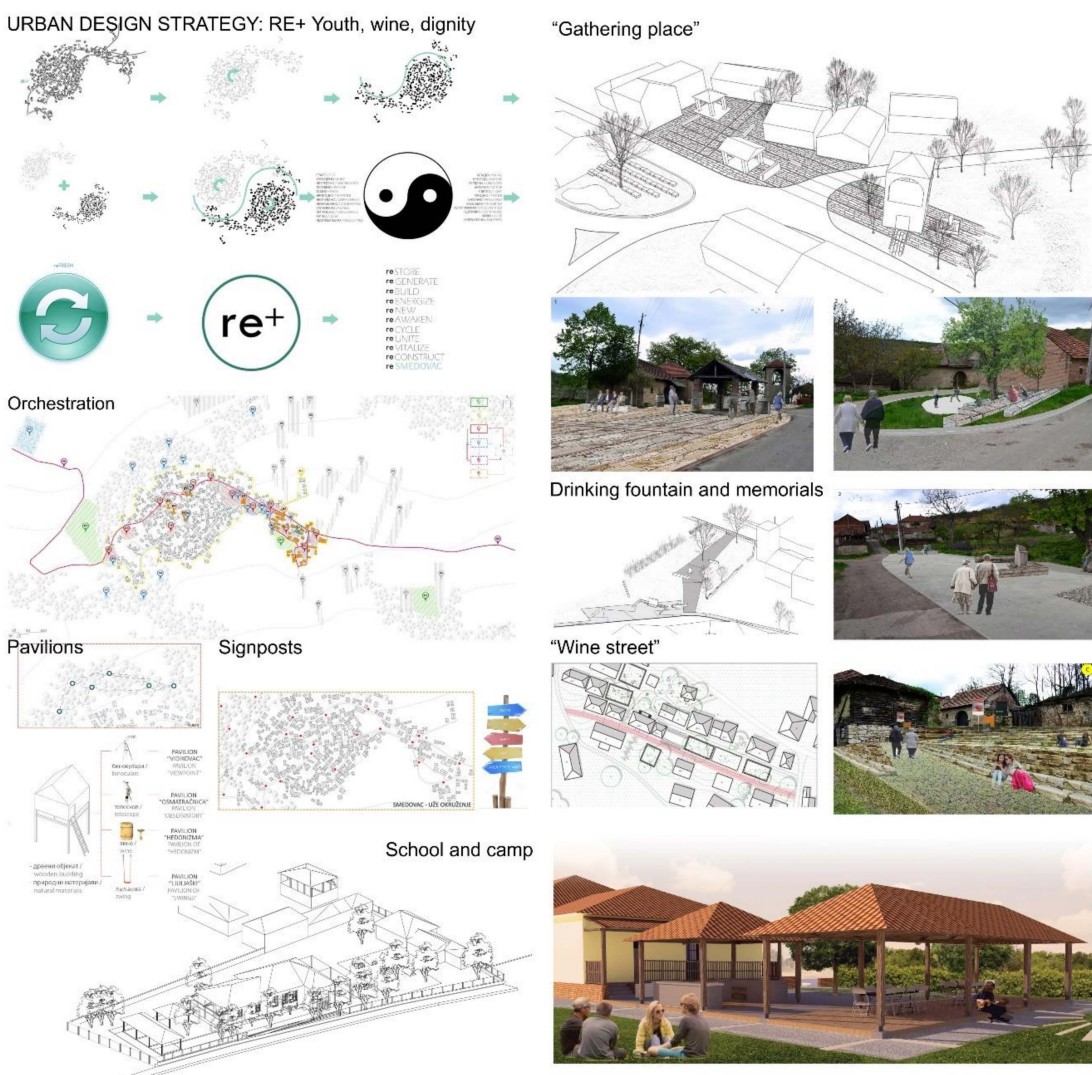

**Figure 12.** Smedovac–Urban design strategy and projects for WCN presentation.

The spatial strategy for WC was based on the recognition that: (a) Conservation plan for WC doesn't exist to provide general guidelines for design; (b) Local community expressed the interest, but at present does not have the capacity to be involved in the management and presentation of WC, (c) Local inhabitants identified many important places as important for improving the use and presentation of WC. The focus of the spatial strategy for presentation of WC of Smedovac followed expressed need to RE-vitalise the village by attracting young people to visit and help future development and presentation of both village and WC. It was assumed that in this way traditional wine-production and use of WC of Smedovac would be RE-activated and dignity of local people supported. In this context strategy called RE+ was aimed to appropriately present village and WC of Smedovac as a place attractive for young people and tourism, it was important to propose a design strategy that will map, use, highlight, and link the existing spatial resources for this purpose. The integral spatial presentation strategy for the village and WC ("Orchestration") was proposed linking individual UD projects of different scales.

- UD Projects for WCN presentation (Figure 12)
  1. "Orchestration"—spatial presentation strategy brings together and harmonise different aspects, actors, UD projects and timelines for their implementation. It applies and integrated view of designing, activating and presenting village, WC and cultural landscape.

2. "School and Camp"—Reconstruction and adaptive use of the abandoned school as well as activation of its surroundings for Camping, is one of the key UD interventions, that aims to trigger future development with minimum investments. It aims to provide temporary, low-cost accommodation as well as the space for lectures, presentations, and workshops for young people and tourists.

3. "Drinking Fountain and memorial"—This design project is located at the strategic point between village and WC aims to highlight local stories and history. It is an important display setting and a stop in walking and guided tours, where also lectures can take place due to new urban furniture.

4. "Gathering Place"—The location consists of several important buildings that tell the story about the village and its viticulture. There are the old bell tower, gathering place building, new bell tower and the monument. By the use of different urban design elements (pavement by use of local materials, lighting, street furniture...), the project aims to enable the variety modes of presentation, while at the same time integrating the space into a unique whole that highlights the place itself as display of traditional rural architecture. Lectures, presentations of local products as well as a variety of activities for tourists and visitors are enabled through the establishment of the small "scene".

5. "Wine Streets"—the main goal of the project is to help tourists and visitors become aware of the variety and richness of the wine-cellars as both rural built heritage, part of viticulture and industrial archaeology. The different materialization of paths is used to stress different relations between the buildings, showcase the traditional use of local materials, and set thus the scene for walking and guiding tours.

6. "Pavilions"—design for different wooden pavilions for different uses ("viewpoint", "observatory", "hedonism", "swings") was proposed and located in the nearby woods—in order to help experience and promote local viticulture, and rural landscape. At the same time, pavilions were supposed to be used as stops in guided and walking tours or for lectures and activities with visitors.

7. "Signposts"—the network of signposts, info-panels and path directions was proposed at different locations in the village and WC as a part of presentation and communication infrastructure.

Integration of Local Knowledge into UD

The level of participation of local community in research for urban design was very high and reflected their interest and enthusiasm for future WC development for both wine-production and tourism. This helped the formation of the complex knowledge base for urban design and WC presentation. It was possible to conceptualize urban design projects as the expression of local people's perception of WC as part of their existential space. Local people perceive WC as an integral part of their existential space because the village and WC are spatially and functionally directly connected. Types of existential space identified by locals as important to be presented, included centers, paths and areas in their complex relations.

## 4. Discussion

Local community knowledge and perception on WCN as existential space was gained through participation process and interpreted through urban design projects on both strategic and individual project level. Specific urban design projects enabled different formats and character of WCN presentation. The overview of the content of local community knowledge, character urban design projects and possibilities for CHS presentation they provide in all WCN locations are presented in Table 2 in order to allow for comparison. The discussion on how participatory urban design can contribute to appropriate heritage presentation focus on main research questions, and critically evaluates WCN PUD project in relation to the principles and objectives of Ename Charter.

**Table 2.** Comparative analysis of cases: WCN PUD knowledge base—urban design projects—heritage presentation potential.

| Local Community Knowledge | | | | Urban Design Projects Reflecting LC Existential Space | | | | CHS Presentation |
|---|---|---|---|---|---|---|---|---|
| Input: Problems | Input: Potentials | Input: Stories | Input: Places | | Theme Function | Location | Form | Formats and Character (Formal/Informal) |
| **Rajac** | | | | **UD Strategy: Re-Connect** | | | | **Formal/Informal** |
| Infrastructure | Cultural heritage | Stories of celebrating and mourning | Scenic viewpoint | 1. | "Belvedere" – scenic viewpoint | Wine cellars,"In between" | Centre and place | Panels and signposts Lectures/Perform./Present Walking and guided tours |
| Lack of gathering places | Wine production(active) | "Hidden paths" between village - WCN | Paths | 2. | "Gathering place" | Wine cellars | Centre and place | Panels and signposts Displays Lectures/Perform./Present Walking and guided tours Activities for visitors |
| | Local initiatives in tourism | | "Popov kladenac" location | 3. | "Popov kladenac"(the priest's well) | Village | Centre and place | Panels and signposts Lectures/Perform./Present Walking and guided tours |
| | | | | 4. | "Places in-between" | Wine cellars | Area and domains | "Living" Displays Activities for visitors |
| | Local identity | "Popov kladenac" (the priest's well) story | Night ambiance /outdoor lighting | 5. | Path from Village to Wine cellars | "In between" | Directions and paths | Panels and signposts Walking and guided tours Activities for visitors |
| | | | | 6. | Path to woods | "In between" | Directions and paths | Panels and signposts Walking and guided tours |
| | | | Local materials/pavement | 7. | Paths through Wine cellars | Wine cellars | Directions and paths | Panels and signposts "Living" Displays Walking and guided toursActivities for visitors |
| **Rogljevo** | | | | **UD Strategy: Sounds & Silence** | | | | **Formal/Informal** |
| Infrastructure | Cultural heritage | Stories of settlements for people and wine | Local materials/pavement | 1. | Path from Village to Wine cellars | "In between" | Directions and paths | Panels and signposts "Living" Displays Walking and guided tours Activities for visitors |
| Lack of gathering places | Wine production(active) | "sounds and silence" | Night ambiance /outdoor lighting | 2. | "Gathering place" | Wine cellars | Centre and place | Panels and signposts Displays Lectures/Perform./Present Walking and guided tours Activities for visitors |
| | Local initiatives in tourism | Historic events | Memorial place | 3. | Memorial drinking fountain | Village | Centre and place | Panels and signposts Displays Walking and guided tours |
| | Local identity | | | 4. | "Places in-between" | Wine cellars | Area and domains | Panels and signposts Walking and guided tours |
| **Štubik** | | | | **UD Strategy: IN Between** | | | | **Formal** |
| Infrastructure Abandoned/ detached from locals | Cultural heritage Recognised as potential by Municipality | / | / | 1. | "Places in-between" | Wine cellars | Area and domains | Panels and signposts Lectures/Perform./Present Activities for visitors |
| | | | | 2. | "Wetland" | Wine cellars | Directions and paths | Panels and signposts Walking and guided tours |

**Table 2.** Comparative analysis of cases: WCN PUD knowledge base—urban design projects—heritage presentation potential.

| Local Community Knowledge | | | | Urban Design Projects Reflecting LC Existential Space | | | | CHS Presentation |
|---|---|---|---|---|---|---|---|---|
| Input: Problems | Input: Potentials | Input: Stories | Input: Places | Theme Function | | Location | Form | Formats and Character (Formal/Informal) |
| Smedovac | | | | UD Strategy: RE + Youth, Wine, Dignity | | | | Formal/Informal |
| Infrastructure | Cultural heritage | Memories of: | School Monument & gathering place | 1. | Orchestration | Whole area | Area and domains | Panels and signposts Displays |
| | | Large village, young people and school | | 2. | School and camp | Wine cellars | Area and domains | Lectures/Perform./Present Activities for visitors |
| Services | Local identity | Wine production | Wine street location | 3. | Drinking fountain and memorial | "In between" | Centre and place | Panels and signposts Lectures/Perform./Present Walking and guided tours |
| Depopulation | Enthusiasm and traditional knowledge | Historic events | Night ambiance /outdoor lighting | 4. | "Gathering place" | Wine cellars | Centre and place | Panels and signposts Lectures/Perform./Present Walking and guided tours Activities for visitors |
| Lack of gathering places | | | Drinking fountain location | 5. | "Wine street" | Wine cellars | Directions and paths | Panels and signposts "living" Displays Walking and guided tours |
| | | | Local materials/ resources | 6. | Pavilions | Whole area | Centre and place | Displays |
| | | | | 7. | Signposts | Whole area | Centre and place | Panels and signposts |

*4.1. How Can Local Community Participation Contribute to Formation of UD Knowledge Base?*

The analysis of the involvement of local communities in the urban design process for WCN presentation reveals several issues. First of all, research reveals that local community is not homogeneous group and that it is therefore important to include a wide spectrum of ACTORS in participatory process, as suggested by previous research [23,24]. Our research showed that the knowledge and perception of WCN vary between actors at municipality and village level, as well as by public and private sector. This is important for guiding priorities and capacities for the development of WCN and influence the approach to WCN presentation through UD.

Secondly, besides involving a variety of actors, the use of different participation FORMS within PUD WCN process, helped gain the different kinds of information to guide urban design strategies and projects. Knowledge gained through questionnaires and surveys enabled identification of key problems and potentials of WCN in general as well as for individual locations. This was used as a basis for UD and presentation strategy formation. But, also, more specific knowledge on (hi)stories and places was gained through walking tours and direct talks with local inhabitants. This knowledge was used for selection of the priority places and guided the character of intervention and presentation. It is this specific knowledge that reflects WCN as an existential place of locals. These findings are also aligned with previous research [25].

Thirdly, the involvement of local communities in different PHASES of UD process confirms to be important for several reasons. During Preparation phase, Municipality representatives helped in building initial PUD project knowledge base by providing strategic documents. The involvement of stakeholders from different sectors (public, private, civic) and levels (municipality/village) during the Field research phase were crucial for acquiring of different kinds of knowledge on space. It revealed differences and similarities among actors, but also enabled portraying each WCN as an existential space of local people. During the Design phase, local people participated as consultants and helped design project development through an iterative process. Finally, Local communities also had an important role as key critics during Presentation and validation phase—when they formally (questionnaire) and informally commented and discussed urban design proposals presented in the exhibition in all WCN locations, as well as in Negotin. In this way they validated the project and suggested possible modifications for more sustainable implementation.

Finally, our research also reveals that *continuity* of community involvement matters. At municipality level, previous positive experiences helped Municipality of Negotin learn about the potentials of collaborative research and showcased the potential of the academic institution to function as link with international institutions and partners. This enabled the establishment of the relations of understanding and trust between the Municipality of Negotin and UBFA as research institution, which made it easier to organize WCN PUD project. At the village community level, continuity of involvement in PUD process has positive effects on empowering rural inhabitants. By involving them in all phases of participatory process, they learned about possibilities to participate and actively guide the future of their CHS.

*4.2. How Can Local Community Knowledge and Perception of WCN as Existential Space Guide Urban Design?*

The analysis of data, gathered through various research techniques, reveals WCN as a part of local people existential space. Through history, they developed different spatial, functional and sociocultural relationships with CHS. The way they perceive and use WCN makes a landscape approach to CHS important; different spatial relations between WCN, village, and in-between and surrounding spaces exist, and were identified as important LOCATIONS in their life-world.

Besides that, local people's knowledge on WCN, as expressed through their stories about of places, helped define specific THEME/FUNCTION of the identified locations that further guided character of urban design.

In addition, in places where direct contact and guided walks with local people were conducted (Rajac, Rogljevo and Smedovac) it was possible to reveal that all FORMS of existential place (center/place, area/domains; directions/paths) are recognized as important and therefore included in spatial presentation strategy and urban design projects. This creates the potential for complex CHS presentation formats to be defined in order to create diverse and memorable touristic experiences for a variety of users.

The results from the WCN PUD project reveal that the inclusion of local people's knowledge and values widens UD knowledge base at strategic and project level. At the strategic level, by: (a) highlighting multi-scalar and diverse spatial, functional and symbolic relationships between people and places, and (b) clarifying interests and capacities for development as precondition for possible presentations strategies that UD should support. At the project level, by: (a) highlighting significant places and paths; (b) revealing hidden uses; and (c) indicating solutions and resources to be used for design.

### 4.3. What Kind of CHS Presentation WCN PUD Project Enables?

Developed on the basis of local people's knowledge and with a goal to support further active use and presentation, WCN PUD projects contribute simultaneously to CHS presentation and creation of memorable and diverse visitors' experiences through physical and functional dimension of urban design.

#### Formal/Informal Presentation

In relation to presentation of rural CHS local people may have several roles to play: (a) *formal*—when they work as formal guides or as service providers; (b) *informal*—when their everyday life and activities in place function as a content of the touristic presentation of CHS. Through these activities they keep WCN CHS heritage "live", and through management of individual WCN they contribute to the quality of CHS environment, as an important part of the tourist experience.

Participation of local communities in the urban design process affects CHS presentation in both direct and indirect ways. Directly—through UD projects that include different presentation formats and enable formal and informal presentation activities and touristic services. Participatory urban design strategy and projects enabled spatially and functionally organized presentation activities, but also revealed local inhabitants' life-world and functional relationships with WCN as a kind of informal CHS presentation. Indirectly, the PUD process revealed certain constraints and limits that affected possibilities to implement specific presentation approaches. For example, in Štubik WCN case, informal presentation is not realistic because local people no longer use their WCN.

#### Presentation Formats

Urban design projects that followed suggestions and knowledge gained by locals, reflect WCN as their existential space in all of its forms (Centre and place, Directions and paths, Area and domains). In that sense, they enabled a variety of presentation formats to be planned (Panels and signposts; Displays; Lectures/Perform/Presentation; Walking and guided tours; Activities for visitors) in Rajac and Rogljevo, and at certain level in Smedovac. Presentation formats anticipated and included in urban design projects were limited to basic communication infrastructure and signposts in the case of Štubik, since its future development scenario is still open and uncertain.

### 4.4. Are There Differences between WCN Locations in Relation to Their Potential for CHS Management and Presentation?

Comparative analysis of different WCN locations shows that specific social and spatial CONTEXT affects possibilities for CHS design, management and presentation. The research reveals that potential for community involvement in the management and presentation

of CHS differs for different locations and can influence the choice and appropriateness of CHS management and presentation model. Several context-related factors have been recognized as important:

- *Location* of WCN in relation to village. Spatial distance from village threats sustainability of WCN in the case of Štubik, while proximity keeps emotional and symbolic ties to WCN alive in the case of Smedovac. When the distance is not too large, functional relations exist (case of Rogljevo and Rajac).
- *Capacity* of the local population to use and manage WCN. The age of population functions as a limiting factor, since it diminishes active use of WCN for wine production(the case of Smedovac).
- *Motivation* of the local population to use and manage WCN. Local people active attitude towards tourism development of WCN has been expressed in Rajac and Rogljevo and works as a motivation factor for active involvement in the presentation of CHS. Surprisingly, although not using WCN for wine production anymore, people in Smedovac have strong emotional connections to place and expressed the will and motivation to support their revitalization and future management, while inhabitants of Štubik stayed passive.

This results in different levels of appropriateness for managing WCN as a living place and suggests that although preferable in rural settings, "Living heritage" management model is not always possible to implement. Relatively good connection between the village and WCN, middle range capacity of high motivation of local populations of Rajac and Rogljevo, make these WCN suitable for "Living heritage" management model, and presentation in accordance to this, but this does not stand for Štubik and Smedovac.

This has implications for conceptualizing UD strategy and projects for CHS presentation. In the case of Rajac and Rogljevo local people can have all aforementioned roles, public spaces were designed to support WCN as a living heritage site in both its environmental and usage dimension. In the case of Smedovac, local people have the knowledge to guide urban design, and may work as guides in presenting their WCN heritage. Public spaces were designed accordingly, mainly focusing on environmental dimension, but envisioning temporal activities to attract visitors. In the case of Štubik, no relation or motivation exists by local people to be included in the presentation. In the context of an uncertain future, UD interventions should focus on connective elements—paths that will be important in any development scenario.

### 4.5. Learning from WCN PUD Project

Based on previous analysis and discussion on WCN PUD Project results, we can now summarize how can participatory urban design contribute to appropriate heritage presentation in Figure 13, that show the relationships between local communities' participation, knowledge base, urban design projects and the potential for WCN presentation.

### 4.6. WCN PUD Project Quality Verification

The quality of the WCN PUD project was verified by both local communities and experts, thus confirming its contribution to cultural heritage theory and practice at national level. In order to verify the general contribution of participatory approach to appropriate presentation, it is important to discuss the WCN PUD project results in relation to Ename Charter principles and objectives.

The Relationship between Ename Charter Principles and WCN PUD Process and Projects

The results presented in Table 3 confirm appropriateness of participatory approach for WCN presentation and showcase how Ename Charter principles may be integrated into both urban design process and projects. They clearly show how involvement of local communities in participatory urban design process contributes to better access and understanding of CHS, enables gaining knowledge from different sources, helps appreciating context and settings, emphasize authenticity, supports sustainability, reflects inclusiveness

and supports research, evaluation and training important to both experts and locals, as basis for appropriate CHS presentation.

**Participatory urban design for cultural heritage site presentation**

**Figure 13.** The relationships between local communities' participation, knowledge base, urban design projects and potential for WCN presentation.

Besides that, the fact that these principles relate to both PUD process and projects confirms validity of understanding urban design as both process and product, and as socio-spatial activity that links people and places, when using it as a tool for CHS presentation.

WCN PUD Projects' Potentials and Limits to Accomplish Ename Charter Objectives of Interpretation and Presentation

The results presented in Table 4 suggest that *potentials* of WCN PUD project to achieve Ename Charter objectives are various and highlight the importance of participatory approach. This relates to almost all objectives. The involvement of local communities from different sectors and levels, in different forms and phases of participatory urban design process directly contributes to the objectives: to facilitate understanding and appreciation of CH; to communicate the meaning; to identify tangible and intangible values in order to safeguard them; it helps understanding and respect for authenticity; encourages inclusiveness in interpretation and contribute to sustainable conservation by directly empowering local people to participate in the future management and presentation of their heritage.

*Limits* of WCN PUD project to achieve Ename Charter objectives also exist. At the first level they relate to participatory process that was limited in time and organization so that some important stakeholders were not possible to include, and some participatory techniques were not possible to apply. Therefore, for better understanding and appreciation, as well as communication of the meaning and respect of authenticity, wider audience should be included in CHS planning and design process through a variety of participatory formats. On the other hand, since due to economic and organizational obstacles the WCN PUD project was developed only at design level, it would be important to evaluate it when implemented in practice. Besides that, rural spatial context set some limits in a way that it is more difficult to inform and organize associated communities to participate in the CHS development process.

Table 3. The relationship between Ename Charter principles and WCN PUD process and projects.

| No. | Principles (Ename) | WCN PUD Process | WCN PUD Projects |
|---|---|---|---|
| 1. | **Access and Understanding** | PUD process helps acquiring local knowledge for better understanding of WCN value and meanings. | PUD projects function as a support for different forms of interpretation and presentation programmes. They also aim to improve the physical condition of public spaces and their access, while at the same time contributing to memorable tourist experiences by revealing, highlighting or amplifying specific features and meanings of WCN as living heritage. Indirectly, they contribute to access and understanding of WCN by enhancing local communities' quality of life and the environment as a prerequisite for protecting the traditional wine production and viticulture protection and presentation. |
| 2. | **Information Sources** | PUD process enabled gaining knowledge from different sources—scientific and living cultural traditions. | PUD projects managed to integrate local oral knowledge on places by interpreting them as elements of existential space, and following relations that exist between local people – WCN and landscapes. In locations where local participation enabled the formation of rich knowledge base and clear focus (Rajac, Rogljevo, Smedovac) it was possible to propose design strategies and project that enables a wide spectrum of presentation formats and infrastructures: panels, displays, lectures, walking and guided tours, places for activities and workshops. |
| 3. | **Context and Setting** | Involvement of local communities in PUD process helped reveal ways they establish their existential space. | Knowledge gained through participation of associated communities provided knowledge on how WCN relates to their wider social, cultural, historical and natural contexts and settings. This was further integrated into spatial strategies and projects of different kinds. Public space projects were designed to enable presentation of both tangible and intangible heritage of local communities. They also recognised the importance of a wider landscape for understanding viticulture and WCN and different kinds of scenic viewpoints were designed to highlight this relation. |
| 4. | **Authenticity** | The basic goal of PUD was to respect traditional social functions, and knowledge was gained through participation process. | Since the starting point of PUD project was to present WCN as living heritage site – protection of WCN fabric and values was of main importance. Formal conservation guidelines were strictly respected. In design projects, all visible interpretive infrastructures were designed as sensitive to the character, setting and the cultural and natural significance of the site. The use of local materials and elements of design was initiated and supported by local people. Locations for On-site concerts, dramatic performances, and other interpretive programmes were carefully planned to protect the significance and physical surroundings of the site. |
| 5. | **Sustainability** | PUD process helped reveal constraints for management and presentation. | PUD strategies and projects were conceptualised with sustainable development perspective in mind and based on specific spatial and social contexts. Presentation of WCN was conceptualised with a goal to support the local life of the place, which by itself becomes part of the presentation of WCN as a living heritage site. |
| 6. | **Inclusiveness** | Meaningful collaboration between academia, and local communities was achieved in PUD process. | Different ways of local communities' participation helped better understanding of traditional rights and interests of property owners and host communities and their integration into urban design projects for presentation of WCN. |
| 7. | **Research, Evaluation, and Training** | PUD project is a continuation of previous collaboration between Negotin Municipality and UBFA. As such, it is an example of continuing research and learning for both partners. | The urban design projects are conceptualised as multifunctional and adaptable, wherever possible in order to be flexible for future revisions or expansion. Proposed projects are modest in their scope, taking into account economic constraints, but they are also based on the precautionary principle in order to be able to reflect and react to effects on the environment. |

**Table 4.** WCN PUD projects' potentials and limits to accomplish Ename Charter objectives.

| No. | Objectives of Interpretation and Presentation (Ename) | WCN PUD Project Potentials | WCN PUD Project Limits |
|---|---|---|---|
| 1. | **Facilitate understanding and appreciation** of cultural heritage sites and foster public awareness and engagement in the need for their protection and conservation | In PUD, different forms of local community participation in research and validation phases fostered public awareness, and contributed to better understanding, appreciation of WCN and future engagement of the local community. | Some additional value of the PUD could have been accomplished if the participation of local communities could have been organized in the form of design charrette. |
| 2. | **Communicate the meaning** of cultural heritage sites to a range of audiences through careful, documented recognition of significance, through accepted scientific and scholarly methods as well as from living cultural traditions | PUD project accomplished this goal through careful planning and realisation of different phases that enabled researchers to acquire different forms and ways of knowing, understanding and valuing WCN, in order to communicate authentic WCN meaning through urban design | PUD managed to test the quality of WCN presentation only as a design project. If any of the projects are delivered in the future, it will be an opportunity to validate the quality of communication of the WCN meaning to a range of audiences. |
| 3. | **Safeguard the tangible and intangible values** of cultural heritage sites in their natural and cultural settings and social contexts | Through different forms of participation in PUD research, different forms of knowledge was acquired that helps safeguard tangible and intangible values of WCN in their natural, social and cultural context. | Local spatial and social contexts, such in the case of Štubik village and WC, affect local communities' involvement in participation through which specific knowledge about tangible and intangible values can be acquired and safeguarded. |
| 4. | **Respect the authenticity** of cultural heritage sites, by communicating the significance of their historic fabric and cultural values and protecting them from the adverse impact of intrusive interpretive infrastructure, visitor pressure, inaccurate or inappropriate interpretation. | Respect for the authenticity was one of the starting points in research that although following experts' conservation guidelines aimed to involve local communities to participate in PUD in the situation when their heritage sites are to be presented. | PUD project time and organisational limits made it impossible to discuss authenticity in more detail with experts from the National Institute for Cultural Heritage. These shortcomings were managed through strict obedience of guidelines and plans (Rajac and Rogljevo) as well as with the application of Institutes methodology for places (Štubik and Smedovac) where studies and plans did not exist. |
| 5. | **Contribute to the sustainable conservation** of cultural heritage sites, through promoting public understanding of, and participation in, ongoing conservation efforts, ensuring long-term maintenance of the interpretive infrastructure and regular review of its interpretive contents. | With a main aim to "bring back hope to local people" PUD main goal was not only to enable quality presentation of WCN to tourists, but to help local recognise their place in management, interpretation and presentation of their heritage. This was achieved both through a UD process in which they were listened and respected and educated, but also through UD projects through which their stories of events and places were revealed and valued. | Wider discussion between stakeholders, not only from local, but also from regional and national level was not realised through PUD – but it would be important to organise it in order to reveal and harmonize different interests in WCN management as national CHS with potential to be WHS. |
| 6. | **Encourage inclusiveness in the interpretation** of cultural heritage sites, by facilitating the involvement of stakeholders and associated communities in the development and implementation of interpretive programmes. | PUD main goal was to encourage inclusiveness in the future management and interpretation of WCN – and therefore involved public, private and civic stakeholders in different forms and phases of the participative urban design process. | Inclusiveness was limited due to the specific rural context in which it is difficult to inform and organise associated communities to participate. This was partially managed through involving local commissioners. |
| 7. | **Develop technical and professional guidelines** for heritage interpretation and presentation, including technologies, research, and training. Such guidelines must be appropriate and sustainable in their social contexts. | This was not a part of PUD. | / |

## 5. Conclusions

In a search for new modalities of social and economic development, rural communities use their natural and cultural heritage for tourism development. Under the new heritage management and tourism development paradigms, the involvement of local communities is recognized as important for sustainable development and presentation of cultural heritage sites. Although community involvement and participation have been researched in

relation to urban and heritage planning and management, little is known about how it can contribute to appropriate presentation of rural, settlement- like heritage sites to tourists and visitors.

Therefore, a representative case of "Wine cellars of Negotin—participatory urban design" (WCN PUD) project was studied in this paper as an example of a successful practice for use of community participation for presentation of CHS through urban design, in Negotin, Serbia. In order to reveal "living" dimensions of WCN that support its sustainability and how they can be integrated into urban design, the study was based on theoretical understanding of urban design as both process and product, and on cultural heritage site as part of local community existential space. The analysis focused on links between the character of community involvement in participatory process, the lay knowledge gained through the process, and urban design solutions for presentation of cultural heritage sites as living places.

The research reveals that participation of different community representatives, in different WCN PUD phases and through different participation formats, helps formation of complex design knowledge base, which, in specific socio-spatial context, affects urban design at both strategic and project levels. This widened knowledge base provides urban design with information that, through recognition of WCN potentials and problems at the municipality level, affects not only design strategies, but also reveals the potential for involvement of local communities in CHS presentation in both formal and informal ways. Besides that, specific knowledge gained at village level, guide urban design solutions by focusing on specific locations, and through providing valuable information that determines the character of their theme/function and form. In that way, it sets the ground for diverse presentation formats, as well as for both formal and informal presentations of the cultural heritage site as both living place and attraction to diverse categories of tourists and visitors.

Nevertheless, our comparative research also reveals that, in relation to potential to develop specific site as living heritage, social and spatial contexts play a significant role. In the case of Štubik, spatial distance plays significant role in separating people from their CHS, while in Smedovac aged population is a limiting factor for the presentation and development of WCN as a living place.

All of this means that the complex structure of stakeholders, as well as diverse and continual involvement of local communities in the development of design solutions, are important factors since they provide a rich knowledge base for development of appropriate design solutions for sustainable CHS presentation in specific rural context.

The implications of these findings are important for both theory and practice. At the theoretical level, the results support theories of community participation that highlight the importance of diversity and continuity in participation process, as well as theories of urban design that conceptualize urban design as both process and product. Besides that, by revealing ways of how local communities' knowledge and values can be integrated and presented through urban design, it contributes to better understanding of why and how local lay knowledge is important for the appropriate presentation of settlement-like rural heritage sites to a variety of visitors.

At the level of practice, this new knowledge and approach should be integrated into public policies in Serbia and elsewhere, if we want to harmonize heritage protection and touristic presentation and develop cultural heritage sites as living places. In words of one of the First principles, defined in "Hoi An Protocols for best conservation practice in Asia" [72]: "The conservation process succeeds when histories are revealed, traditions revived and meanings recovered in a palimpsest of knowledge", resulting in a life-enhancing space for both inhabitants and visitors.

Further studies, offering analysis of more complex participation and collaboration mechanisms between different stakeholders, responses and experience from local communities, or reflections on the implemented participatory urban design projects, would provide useful contributions to this research.

**Author Contributions:** Conceptualization, J.Ž. and Z.Đ.; Methodology, J.Ž. and Z.Đ.; Investigation, Z.Đ., J.Ž., P.J; Writing—original draft preparation, J.Ž. and Z.Đ.; Writing—review and editing, J.Ž, Z.Đ, U.R., K.L., P.J.; Visualization J.Ž. and P.J. All authors have read and agreed to the published version of the manuscript.

**Funding:** Research received no external funding.

**Institutional Review Board Statement:** Not applicable.

**Informed Consent Statement:** Not applicable.

**Data Availability Statement:** Data can be made available on request.

**Acknowledgments:** We would like to thank the anonymous reviewers for their useful comments and suggestions; the Academic editors for their valuable work; 2014/15 generation of students from the UBFA "Participatory Urban Design Studio" course, Municipality of Negotin and other participants in WCN PUD project.

**Conflicts of Interest:** The authors declare no conflict of interest.

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
