# Peer review of "Participatory Urban Design for Touristic Presentation of Cultural Heritage Sites: The Case of Negotinske Pivnice (Wine Cellars) in Serbia"

_sustainability, doi:10.3390/su131810039_

Round 1

Reviewer 1 Report

The text presented for evaluation represents the highest scientific level. All discovery criteria have been met. The text has a clear research purpose, a large empirical base, and is based on a good knowledge of the problem. The bibliography and its use is excellent. The undertaken problem is important not only scientifically but also socially. The presented research can gain a great international influence, both in the world of science and in the transformations of rural centers with large cultural traditions.

Reviewer 2 Report

Dear esteemed authors,

Thank you very much for this brilliant active-research and very interesting manuscript.

I enjoyed reading your co-authored paper and really appreciated the multidisciplinary literature review. Another strength point is the case-study structure and analysis and above all the diverse and contexualised findings. My only two comments are:

-to try to cut the redundant text;

-to explain in a footnote the different incorporated disciplines/professional backgrounds in the research 

Seen the importance of the research and the very well-structured and detailed process, I would suggest submitting two articles. The first would anticipate the theoretical framework, research questions, methodology and results while the second would provide evidence-based results (the four case-studies). This is of course a suggestion only and it is up to the authors to decide.  
